# AN INFORMATION THEORETIC APPROACH TO DISTRIBUTED REPRESENTATION LEARNING

## ABSTRACT

The problem of distributed representation learning is one in which multiple sources of information $X_1, \ldots, X_K$ are processed separately so as to extract useful information about some statistically correlated ground truth $Y$. We investigate this problem from information-theoretic grounds. For both discrete memoryless (DM) and memoryless vector Gaussian models, we establish fundamental limits of learning in terms of optimal tradeoffs between *relevance* and *complexity*. We also develop a variational bound on the optimal tradeoff that generalizes the evidence lower bound (ELBO) to the distributed setting. Furthermore, we provide a variational inference type algorithm that allows to compute this bound and in which the mappings are parametrized by neural networks and the bound approximated by Markov sampling and optimized with stochastic gradient descent. Experimental results on synthetic and real datasets are provided to support the efficiency of the approaches and algorithms which we develop in this paper.

## 1 INTRODUCTION

Let a measurable variable $X \in \mathcal{X}$ and a target variable $Y \in \mathcal{Y}$ with unknown joint distribution $P_{X,Y}$ be given. In the classic problem of statistical learning, one wishes to infer an accurate predictor of the target variable $Y \in \mathcal{Y}$ based on observed realizations of $X \in \mathcal{X}$. That is, for a given class $\mathcal{F}$ of admissible predictors $\phi : \mathcal{X} \to \hat{\mathcal{Y}}$ and an additive loss function $\ell : \mathcal{Y} \to \hat{\mathcal{Y}}$ that measures discrepancies between true values and their estimated fits, one aims at finding the mapping $\phi^{\star} \in \mathcal{F}$ that minimizes the expected risk

$$\mathcal{C}_{P_{X,Y}}(\phi, \ell) = \mathbb{E}_{P_{X,Y}}[\ell(Y, \phi(X))]. \tag{1}$$

Because the joint distribution $P_{X,Y}$ is unknown, in practice the risk equation 1 (also called *population risk*) cannot be computed directly; and, in the standard approach, one usually resorts to choosing the predictor with minimal risk on a training dataset consisting of $n$ labeled samples $\{(x_i, y_i)\}_{i=1}^{n}$ that are drawn independently from the unknown joint distribution $P_{X,Y}$. Also, it is important to restrict the set $\mathcal{F}$ of admissible predictors to a low-complexity class to prevent overfitting. This leads to the abstract inference problem shown in Figure 1.

In this paper, we study a generalization of this problem in which the prediction is to be performed in a distributed manner. The model is shown in Figure 2. Here, the prediction of the target variable $Y \in \mathcal{Y}$ is to be performed on the basis of samples of statistically correlated random variables $(X_1, \ldots, X_K)$ that are observed each at a distinct predictor. We investigate this problem in the case in which the loss function $\ell(\cdot)$ is the logarithmic-loss fidelity measure, given by

$$\ell_{\log}(y, \hat{y}) = \log\left(\frac{1}{\hat{y}(y)}\right) \tag{2}$$

where $\hat{y}(\cdot)$ designates a probability distribution on $\mathcal{Y}$ and $\hat{y}(y)$ is the value of this distribution evaluated for the outcome $y \in \mathcal{Y}$. The choice of a 'good" loss function is often controversial in statistical learning theory, and although a complete and rigorous justification of the usage of logarithmic loss as a fidelity measure in learning theory is still awaited, partial explanations appeared in Jiao et al. (2015) and, especially in Painsky and Wornell (2018) where it is shown that, for binary classification problems, by minimizing the logarithmic-loss one actually minimizes an upper bound to any choice of loss function that is smooth, proper (i.e., unbiased and Fisher consistent) and convex. Also, we constrain the complexity of the predictors by using mutual information as a regularizer term. This is inline with recent works Xu and Raginsky (2017); Russo and Zou (2015) that show that the generalization error can be upper-bounded using the mutual information between the input dataset and the output of the predictor – see also Bousquet and Elisseeff (2002); Shalev-Shwartz et al. (2010) where the *stability* of an algorithm is controlled by constraining the mutual information between its input and output.

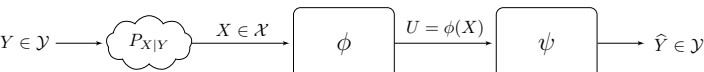

**Figure 1:** An abstract inference model for learning.

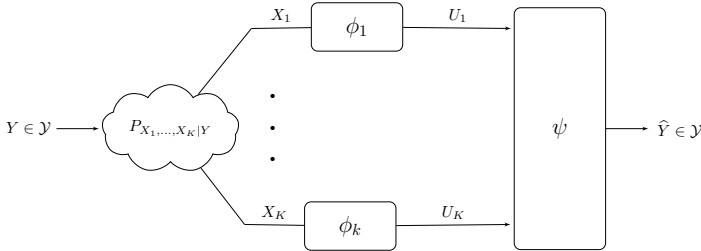

**Figure 2:** A model for distributed, e.g., multi-view, learning.

## 1.1 AN EXAMPLE: MULTI-VIEW LEARNING

In many data analytics problems, data is collected from various sources of information or feature extractors; and is intrinsically *heterogeneous*. For example, an image can be identified by its color or texture features; and a document may contain text and images. Conventional machine learning approaches concatenate all available data into one big row vector (or matrix) on which a suitable algorithm is then applied. Treating different observations as a single source might cause overfitting and is not physically meaningful because each group of data may have different statistical properties. Alternatively, one may partition the data into groups according to samples homogeneity, and each group of data be regarded as a separate *view*. This paradigm, termed *multi-view learning* Xu et al. (2013), has received growing interest; and various algorithms exist, sometimes under references such as *co-training* Blum and Mitchell (1998); Dhillon et al. (2011); Kumar and Daumé (2011); Gönen and Alpaydın (2011), *multiple kernel learning* Gönen and Alpaydın (2011) and *subspace learning* Jia et al. (2010). By using distinct encoder mappings to represent distinct groups of data, and jointly optimizing over all mappings to remove redundancy, multiview learning offers a degree of flexibility that is not only desirable in practice but is likely to result in better learning capability. Actually, as shown in Vapnik (2013), local learning algorithms produce less errors than global ones. Viewing the problem as that of function approximation, the intuition is that it is usually non-easy to find a unique function that holds good predictability properties in the entire data space.

## 1.2 INFORMAL SUMMARY OF RESULTS

In this paper, first we characterize the optimal tradeoff between relevance and complexity for the distributed learning model of Figure 2 for both discrete memoryless (DM) and memoryless vector Gaussian models. While the result for the discrete data model (Theorem 1) is not difficult to establish using connections with Courtade and Weissman (2014, Appendix B) which we explicit here, the result for the multivariate Gaussian data model (Theorem 2), which provides a sharp analytic characterization of optimal tradeoffs, is new and non-trivial (the proof of the converse part is not straightforward and was missing before this work in both theoretic learning and information theoretic communities including in the scalar case). Second, we develop a variational bound on the optimal tradeoff that can be seen as a generalization of the ELBO and the $\beta$-VAE criteria Higgins et al. (2016) to the distributed setting. Furthermore, for both DM and Gaussian models, we also provide a variational inference type algorithm which is parametrized by neural networks and allows to compute the developed variational bound when the data distribution is not known. Specifically, the main contributions of this paper are:

- In Section 3.2, we find an explicit analytic characterization of optimal tradeoffs between relevance and complexity for the memoryless vector Gaussian model. The result generalizes the Gaussian Information Bottleneck method of Globerson and Tishby (2004); Chechik et al. (Feb. 2005) to the distributed learning scenario.

- In Section 3.3, we study the problem of maximizing relevance under a constraint on the sum complexity for which we establish a variational bound which generalizes the ELBO and the $\beta$-VAE criteria to the distributed setting.

- Section 3.4 is algorithmic-oriented. We develop a variational inference type algorithm which enables to compute the bound. This algorithm is obtained by parametrizing the encoders, the decoder, and the prior distributions via DNNs and using Monte-Carlo sampling. Also, it makes usage of Kingma et

al.'s re-parametrization trick Kingma and Welling (2013) and can be seen as a generalization of the variational information bottleneck algorithm in Alemi et al. (2017) to the distributed setting.

- Section 4 contains some experimental results on real datasets which show the efficiency of the approaches and algorithms that we develop in this paper.

Most relevant to this paper is the single-encoder Information Bottleneck (IB) method of Tishby et al. (1999) which readily and elegantly captures the above mentioned viewpoint of seeking the right balance between data fit and generalization by using the mutual information both as a cost function and as a regularizer term. Thus, the results of this paper can be seen as a generalization of those of Tishby et al. (1999) for the DM model and Globerson and Tishby (2004); Chechik et al. (Feb. 2005) for the Gaussian model to the distributed learning setting.

**Remark:** Due to space constraints, the proofs of the results of this paper are deferred to the appendices section, which also contains additional experimental results.

## 1.3 NOTATION

Throughout, upper case letters denote random variables, e.g., X; lower case letters denote realizations of random variables, e.g., $x$; and calligraphic letters denote sets, e.g., $\mathcal{X}$. The cardinality of a set is denoted by $|\mathcal{X}|$. For a random variable $X$ with probability mass function (pmf) $P_X$, we use $P_X(x) = p(x)$, $x \in \mathcal{X}$ for short. Boldface upper case letters denote vectors or matrices, e.g., $\mathbf{X}$, where context should make the distinction clear. For random variables $(X_1, X_2, \ldots)$ and a set of integers $\mathcal{K} \subseteq \mathbb{N}$, $X_\mathcal{K}$ denotes the set of random variables with indices in the set $\mathcal{K}$, i.e., $X_\mathcal{K} = \{X_k : k \in \mathcal{K}\}$. If $\mathcal{K} = \emptyset$, $X_\mathcal{K} = \emptyset$. For $k \in \mathcal{K}$ we let $X_{\mathcal{K}/k} = (X_1, \ldots, X_{k-1}, X_{k+1}, \ldots, X_K)$, and assume that $X_0 = X_{K+1} = \emptyset$. Also, for zero-mean random vectors $\mathbf{X}$ and $\mathbf{Y}$, the quantities $\mathbf{\Sigma_x}$, $\mathbf{\Sigma_{x,y}}$ and $\mathbf{\Sigma_{x|y}}$ denote respectively the covariance matrix of the vector $\mathbf{X}$, the covariance matric of vector $(\mathbf{X}, \mathbf{Y})$ and the conditional covariance matrix of $\mathbf{X}$, conditionally on $\mathbf{Y}$. Finally, for two probability measures $P_X$ and $Q_X$ on the random variable $X \in \mathcal{X}$, the relative entropy or Kullback-Leibler divergence is denoted as $D_{\mathrm{KL}}(P_X \| Q_X)$.

## 2 FORMAL PROBLEM FORMULATION

Let $K \geq 2$ and $(X_1, \ldots, X_K, Y)$ be a tuple of random variables with a given joint probability mass function (pmf) $P_{X_1, \ldots, X_K, Y}(x_1, \ldots, x_K, y)$ for $(x_1, \ldots, x_K) \in \mathcal{X}_1 \times \ldots \times \mathcal{X}_K$ and $y \in \mathcal{Y}$, where $\mathcal{X}_k$ designates the alphabet of $X_k$ and $\mathcal{Y}$ that of $Y$. Throughout, we assume that the Markov chain

$$X_k \; \multimap \; Y \; \multimap \; X_{\mathcal{K}/k} \tag{3}$$

holds for all $k \in \mathcal{K}$. That is, the joint pmf factorizes as

$$P_{X_1, \ldots, X_K, Y}(x_1, \ldots, x_K, y) = P_Y(y) \prod_{k=1}^{K} P_{X_k|Y}(x_k|y). \tag{4}$$

The variable $Y$ is a target variable; and we seek to characterize how accurate it can be predicted from a measurable random vector $(X_1, \ldots, X_K)$ when the components of this vector are processed separately, each by a distinct encoder. More specifically, let $\{(X_{1,i}, \ldots, X_{K,i}, Y_i)\}_{i=1}^n$ be a collection of $n$ independent copies of $(X_1, \ldots, X_K, Y)$. Encoder $k \in \mathcal{K}$ only observes the sequence $X_k^n$; and generates a description $J_k = \phi_k(X_k^n)$ according to some mapping

$$\phi_k : \mathcal{X}_k^n \to \mathcal{M}_k^{(n)}, \tag{5}$$

where $\mathcal{M}_k^{(n)}$ is an arbitrary set of descriptions. The range of allowable description sets will be specified below. A decoder $\psi(\cdot)$ collects all descriptions $J_\mathcal{K} = (J_1, \ldots, J_K)$ and returns an estimate $\hat{Y}^n$ of $Y^n$ as

$$\psi : \mathcal{M}_1^{(n)} \times \ldots \times \mathcal{M}_K^{(n)} \to \hat{\mathcal{Y}}^n. \tag{6}$$

The relevance of the estimation $\hat{Y}^n$ is measured in terms of the *relevance*, defined here as the information that the descriptions $\phi_1(X_1^n), \ldots, \phi_K(X_K^n)$ *collectively* preserve about $Y^n$, as measured by Shannon mutual information [1]

$$\Delta^{(n)}(P_{X_\mathcal{K}, Y}) = \frac{1}{n} \sum_{y^n, x_1^n, \ldots, x_K^n} P(y^n) \prod_{k=1}^{K} P(x_k^n|y^n) \log \frac{P(y^n, \psi(\phi_1(x_1^n), \ldots, \phi_K(x_K^n)))}{P(y^n)P(\psi(\phi_1(x_1^n), \ldots, \phi_K(x_K^n)))}$$

$$:= \frac{1}{n} I_{P_{X_\mathcal{K}, Y}}(Y^n; \hat{Y}^n), \tag{7}$$

---

[1] Alternatively, the relevance could be defined in a more operational manner by the average logarithmic loss distortion or error $\mathbb{E}_{P_{X_\mathcal{K}, Y}}[\ell_{\log}(Y^n, \hat{Y}^n)] = H(Y^n|\hat{Y}^n)$.

where $\hat{Y}^n = \psi(\phi_1(X_1^n), \ldots, \phi_K(X_K^n))$ and the subscript $P_{X_{\mathcal{K}},Y}$ indicates that the mutual information is computed under the joint distribution $P_{X_{\mathcal{K}},Y}$.

There are various ways to control the *complexity* of the encoding functions $\{\phi_k\}_{k=1}^K$. In this paper, we do so by restricting their ranges. This is known as *minimum description length* complexity measure Hinton and van Camp (1993). Specifically, the mapping $\phi_k(\cdot)$ at Encoder $k \in \mathcal{K}$ needs to satisfy

$$R_k \geq \frac{1}{n} \log |\phi_k(X_k^n)| \quad \text{for all} \quad X_k^n \in \mathcal{X}_k^n. \tag{8}$$

**Definition 1** *A tuple $(\Delta, R_1, \ldots, R_K)$ is said to be achievable if there exists an integer $n$, a family of encoding mappings $\{\phi_k\}_{k=1}^K$ and a decoder mapping $\psi$ such that*

$$\Delta \leq \frac{1}{n} I_{P_{X_{\mathcal{K}},Y}}\left(Y^n; \psi(\phi_1(X_1^n), \ldots, \phi_K(X_K^n))\right) \tag{9}$$

$$R_k \geq \frac{1}{n} \log |\phi_k(X_k^n)| \quad \text{for all} \quad k \in \mathcal{K}. \tag{10}$$

*The relevance-complexity region $\mathcal{IR}_{\text{DIB}}$ is given by the closure of all achievable tuples $(\Delta, R_1, \ldots, R_K)$.*

In some cases, for given $R_{\mathcal{K}} = (R_1, \ldots, R_K)$, for the ease of the exposition we will be content with the relevance-complexity function $\Delta(R_{\mathcal{K}}, P_{X_{\mathcal{K}},Y})$ defined as

$$\Delta(R_{\mathcal{K}}, P_{X_{\mathcal{K}},Y}) = \max_{\{\phi_k\}_{k=1}^K, \psi} \Delta^{(n)}(P_{X_{\mathcal{K}},Y}) \tag{11}$$

where the maximization is subjected to equation 8.

# 3 MAIN RESULTS

## 3.1 DISCRETE MEMORYLESS DATA MODEL

The following theorem (the proof of which can be found in the appendices section) provides a computable characterization of the relevance-complexity region $\mathcal{IR}_{\text{DIB}}$. The result can be seen as a generalization of Tishby et al. Tishby et al. (1999) single encoder IB to the distributed learning model with $K$ encoders.

**Theorem 1** *The relevance-complexity region $\mathcal{IR}_{\text{DIB}}$ of the distributed learning problem with $P_{X_{\mathcal{K}},Y}$ for which the Markov chain equation 3 holds is given by the union of all tuples $(\Delta, R_1, \ldots, R_K) \in \mathbb{R}_+^{K+1}$ that satisfy for all $\mathcal{S} \subseteq \mathcal{K}$,*

$$\Delta \leq \sum_{k \in \mathcal{S}} [R_k - I(X_k; U_k|Y,T)] + I(Y; U_{\mathcal{S}^c}|T), \tag{12}$$

*for some set of pmfs $\mathbf{P} := \{P_{U_k|X_k,T}, \ldots, P_{U_k|X_k,T}, P_T\}$ with joint distribution of the form*

$$P_T(t) P_Y(y) \prod_{k=1}^K P_{X_k|Y}(x_k|y) \prod_{k=1}^K P_{U_k|X_k,T}(u_k|x_k,t). \tag{13}$$

**Remark 1** *In Theorem 1, the random variable $T$ stands for a convexification of the region, i.e., convex combination of achievable relevance-complexity tuples is itself achievable. For given $T = t$, the result of Theorem1 comprises the optimization over $K$ conditional distributions $\{P_{U_K|X_k,t}\}$. For $k \in \mathcal{K}$, the conditional distribution $P_{U_K|X_k,t}$ represents a stochastic encoding of the feature $X_k$ into a latent variable $U_k$. Intuitively, the latent variable $U_k$ should capture all relevant information about $Y$ that is contained in $X_k$ and non redundant with those carried out by $\{U_i\}_{i \neq k}$. The requirement of non-redundancy is mandated by the need to operate at the minimum possible complexity at which a desired relevance level is achievable (recall that minimum complexity, as expressed by algorithm's input-output mutual information, translates directly into a better generalization capability). Collectively, however, the set of all latent variables $(U_1, \ldots, U_K)$ should be expressive enough to reproduce the target variable $Y$ to within the desired relevance level.*

**Remark 2** *Like for the single-encoder IB problem of Tishby et al. (1999) and an increasing number of works that followed, including Courtade and Weissman (2014, Section III-F), our approach here is asymptotic. In addition to that it leads to an exact characterization, the result also readily provides a lower bound on the performance in the non-asymptotic (e.g., one shot) setting. For the latter setting known approaches (e.g., the functional representation lemma of Li and El Gamal (2018)) would lead to only non-matching inner and outer bounds on the region of optimal tradeoff pairs, as this is the case even for the single encoder case Li et al. (2018).*

## 3.2 Memoryless vector Gaussian data model

We now turn to a continuous-alphabet setting. Here, $(\mathbf{X}_1, \ldots, \mathbf{X}_K, \mathbf{Y})$ is a zero-mean Gaussian random vector such that

$$\mathbf{X}_k = \mathbf{H}_k \mathbf{Y} + \mathbf{N}_k \quad \text{for all} \quad k \in \mathcal{K}, \tag{14}$$

where $\mathbf{H}_k \in \mathbb{C}^{n_k \times n_y}$ models the linear model connecting the target variable $\mathbf{Y} \in \mathbb{C}^{n_y}$ to the observation at encoder $k$, and $\mathbf{N}_k \in \mathbb{C}^{n_k}$, $k = 1, \ldots, K$, is the noise vector at encoder $k$, assumed to be Gaussian with zero-mean and covariance matrix $\mathbf{\Sigma}_k$, and independent from all other noises and the target variable $\mathbf{Y}$. We denote by $\mathbf{\Sigma}_\mathbf{y}$ the covariance matrix of of the target vector $\mathbf{Y} \in \mathbb{C}^{n_y}$.

For this model, we find an explicit analytic characterization of optimal tradeoffs between relevance and complexity. The proof relies on deriving an outer bound on the region described by equation 12, and showing that it is achievable with Gaussian distribution, with *no* time-sharing. In doing so, we use techniques that rely on the de Bruijn identity and the properties of Fisher information and minimum mean square error (MMSE).

**Theorem 2** *The relevance-complexity region $\mathcal{IR}_{\mathrm{DIB}}^{\mathrm{G}}$ for the vector Gaussian model is given by the union of all tuples $(\Delta, R_1, \ldots, R_L)$ that satisfy for all $\mathcal{S} \subseteq \mathcal{K}$*

$$\Delta \leq \left[ R_k + \log \left| \mathbf{I} - \mathbf{\Sigma}_k^{1/2} \mathbf{\Omega}_k \mathbf{\Sigma}_k^{1/2} \right| \right] + \log \left| \sum_{k \in \mathcal{S}^c} \mathbf{\Sigma}_\mathbf{y}^{1/2} \mathbf{H}_k^\dagger \mathbf{\Omega}_k \mathbf{H}_k \mathbf{\Sigma}_\mathbf{y}^{1/2} + \mathbf{I} \right|,$$

*for some $\mathbf{0} \preceq \mathbf{\Omega}_k \preceq \mathbf{\Sigma}_k^{-1}$.*

**Proof:** The proof of the direct part follows by evaluating the region of Theorem 1, which can be extended to the case of continuous alphabets using standard discretization (quantization) arguments, with the choices $T = \emptyset$ and $p(\mathbf{u}_k | \mathbf{x}_k, t) = \mathcal{CN}(\mathbf{x}_k, \mathbf{\Sigma}_k^{1/2}(\mathbf{\Omega}_k - \mathbf{I})\mathbf{\Sigma}_k^{1/2})$. The main contribution in the proof is that of the converse part. This proof is technical and rather lengthy and, for this reason, is deferred to the appendices section.

In the special case in which $K = 1$, the result of Theorem 2 recovers that by Globerson and Tishby (2004) (see also Chechik et al. (Feb. 2005)) which establishes the optimal relevance-complexity tradeoff of the single-encoder Gaussian IB problem.

## 3.3 A variational bound

In this section, we consider the problem of learning encoders- and decoder mappings that maximize the relevance level for a given (fixed) complexity level, i.e., those that perform at the vicinity of the boundary of the region $\mathcal{IR}_{\mathrm{DIB}}$. First, we derive a parametrization of the relevance-complexity region; and, then, we develop a variational bound which expresses the optimal encoders' and decoder mappings as the solution to an optimization problem – (an algorithm for solving this problem in the case of unknown distributions is given in the next section).

Let $R_{\mathrm{sum}} := \sum_{k=1}^{K} R_k$. Also, let $\mathcal{IR}_{\mathrm{DIB}}^{\mathrm{sum}}$ denote the region of achievable (relevance, sum-complexity) pairs,

$$\mathcal{IR}_{\mathrm{DIB}}^{\mathrm{sum}} := \Big\{ (\Delta, R_{\mathrm{sum}}) \in \mathbb{R}_+^2 : \exists (R_1, \ldots, R_K) \in \mathbb{R}_+^K \text{ s.t.}$$

$$(\Delta, R_1, \ldots, R_K) \in \mathcal{IR}_{\mathrm{DIB}} \text{ and } \sum_{k=1}^{K} R_k = R_{\mathrm{sum}} \Big\}.$$

**Proposition 1** *The relevance-complexity region under sum-complexity constraint $\mathcal{RI}_{\mathrm{DIB}}^{\mathrm{sum}}$ is given by the convex-hull of all tuples $(\Delta, R_{\mathrm{sum}}) \in \mathbb{R}_+^2$ satisfying $\Delta \leq \Delta(R_{\mathrm{sum}}, P_{X_{\mathcal{K}}, Y})$ where*

$$\Delta(R_{\mathrm{sum}}, P_{X_{\mathcal{K}}, Y}) = \max_{\mathbf{P}} \min \left\{ I(Y; U_{\mathcal{K}}), R_{\mathrm{sum}} - \sum_{k=1}^{K} I(X_k; U_k | Y) \right\}, \tag{15}$$

*and where the maximization is over the set of pmfs $\mathbf{P} := \{P_{U_1 | X_1}, \ldots, P_{U_K | X_K}\}$ such that the joint pmf factorizes as $p_Y(y) \prod_{k=1}^{K} p_{X_k | Y}(x_k | y) \prod_{k=1}^{K} p_{U_k | X_k}(u_k | x_k)$.*

The next proposition provides a characterization of the pairs $(\Delta, R_{\mathrm{sum}})$ that lie on the boundary of $\mathcal{RI}_{\mathrm{DIB}}^{\mathrm{sum}}$ in terms of a nonnegative parameter $s \geq 0$.

**Proposition 2** *For every pair $(\Delta, R_{\mathrm{sum}}) \in \mathbb{R}_+^2$ that lies on the boundary of the relevance-complexity region $\mathcal{RI}_{\mathrm{DIB}}^{\mathrm{sum}}$ there exist $s \geq 0$ such that $(\Delta, R_{\mathrm{sum}}) = (\Delta_s, R_s)$, where*

$$\Delta_s = \frac{1}{(1+s)} \left[ (1 + sK)H(Y) + sR_s + \max_{\mathbf{P}} \mathcal{L}_s(\mathbf{P}) \right], \tag{16}$$

$$R_s = I(Y; U_{\mathcal{K}}^*) + \sum_{k=1}^{K} [I(X_k; U_k^*) - I(Y; U_k^*)], \tag{17}$$

*and $\mathbf{P}^*$ is the set of conditional pmfs $\mathbf{P}$ that maximize the cost function*

$$\mathcal{L}_s(\mathbf{P}) := -H(Y|U_{\mathcal{K}}) - s \sum_{k=1}^{K} [H(Y|U_k) + I(X_k; U_k)]. \tag{18}$$

Using Proposition 2 it is clear that the encoders $\{P_{U_k|X_k}\}_{k \in \mathcal{K}}$ that achieve the relevance-complexity pair $(\Delta_s, R_s)$ can be computed by maximizing the regularized cost equation 18 for the corresponding value of $s \geq 0$. The corresponding optimal decoder $P_{Y|U_{\mathcal{K}}}$ for these encoders can be found as in equation **??**. Different relevance-complexity pairs $(\Delta_s, R_s)$ on the boundary of $\mathcal{IR}_{\mathrm{DIB}}^{\mathrm{sum}}$ and encoders- and decoder mappings that achieve it can be found by solving equation 18 for different values of $s \geq 0$ and then evaluating equation 16 and equation 17 for the obtained solution.

The optimization of equation 18 generally requires to compute marginal distributions involving the descriptions $U_1, \ldots, U_K$, an aspect which can be non-easy computationally costly. To overcome this limitation, in the following we derive a tight variational bound on $\mathcal{L}_s(\mathbf{P})$ which lower bounds the DIB cost function with respect to some arbitrary distributions. Let us consider the arbitrary decoder $Q_{Y|U_1,\ldots,U_K}(y|u_1,\ldots,u_K)$ for $y \in \mathcal{Y}$, $u_1 \in \mathcal{U}_1, \ldots, u_K \in \mathcal{U}_K$, the $K$ decoders $Q_{Y|U_k}(y|u_k)$ for $k \in \mathcal{K}$ for $y \in \mathcal{Y}$, $u_k \in \mathcal{U}_k$, and latent variable priors $Q_{U_k}(u_k)$, $k \in \mathcal{K}$, $u_k \in \mathcal{U}_k$. For short, we denote

$$\mathbf{Q} := \{Q_{Y|U_1,\ldots,U_K}, Q_{Y|U_1}, \ldots, Q_{Y|U_K}, Q_{U_1}, \ldots, Q_{U_K}\}.$$

Let us define the *variational DIB cost function* $\mathcal{L}_s^{\mathrm{VB}}(\mathbf{P}, \mathbf{Q})$ as

$$\mathcal{L}_s^{\mathrm{VB}}(\mathbf{P}, \mathbf{Q}) := \underbrace{\mathbb{E}[\log Q_{Y|U_{\mathcal{K}}}(Y|U_{\mathcal{K}})]}_{\text{av. logarithmic-loss}} + s \sum_{k=1}^{K} \underbrace{\left( \mathbb{E}[\log Q_{Y|U_k}(Y|U_k)] - D_{\mathrm{KL}}(P_{U_k|X_k}\|Q_{U_k}) \right)}_{\text{regularizer}}. \tag{19}$$

The following lemma states that $\mathcal{L}_s^{\mathrm{VB}}(\mathbf{P}, \mathbf{Q})$ is a lower bound to $\mathcal{L}_s(\mathbf{P})$ for all distributions $\mathbf{Q}$.

**Lemma 1** *For fixed pmfs $\mathbf{P}$, we have*

$$\mathcal{L}_s(\mathbf{P}) \geq \mathcal{L}_s^{\mathrm{VB}}(\mathbf{P}, \mathbf{Q}), \qquad \text{for all pmfs } \mathbf{Q}. \tag{20}$$

*In addition, there exists a unique $\mathbf{Q}$ that achieves the maximum $\max_{\mathbf{Q}} \mathcal{L}_s^{\mathrm{VB}}(\mathbf{P}, \mathbf{Q}) = \mathcal{L}_s(\mathbf{P})$, and is given by*

$$Q_{U_k}^* = P_{U_k}, \qquad Q_{Y|U_k}^* = P_{Y|U_k}, \quad k = 1, \ldots, K, \tag{21}$$

$$Q_{Y|U_1,\ldots,U_k}^* = P_{Y|U_1,\ldots,U_K}, \tag{22}$$

*where $P_{U_k}$, $P_{Y|U_k}$ and $P_{Y|U_1,\ldots,U_K}$ are computed from the pmfs $\mathbf{P}$.* $\qquad\square$

Using the above, the optimization in equation 16 can be written in terms of the variational DIB cost function as

$$\max_{\mathbf{P}} \mathcal{L}_s(\mathbf{P}) = \max_{\mathbf{P}} \max_{\mathbf{Q}} \mathcal{L}_s^{\mathrm{VB}}(\mathbf{P}, \mathbf{Q}). \tag{23}$$

We close this section by noting that the cost function equation 19 can be seen as a generalization of the evidence lower bound (ELBO) as given in Rezende et al. (2014); Kingma and Welling (2013) for the single-encoder learning to the distributed setting. Also, in the specific case in which $Y = (X_1, \ldots, X_K)$ the bound generalizes the ELBO used for VAEs to the case of an arbitrary number of encoders.

### 3.4 CASE OF UNKNOWN DISTRIBUTIONS: VARIATIONAL DISTRIBUTED IB ALGORITHM

In practice only a set of training samples $\{(X_{1,i}, \ldots, X_{K,i}, Y_i)\}_{i=1}^{n}$ are available. In this section, we provide a method to optimize equation 23 in this case by parametrizing the encoding and decoding distributions that are to optimize using a family of distributions whose parameters are determined by Deep Neural networks (DNNs). This allows us to formulate equation 23 in terms of the DNN parameters and optimize it by using the reparametrization trick Kingma and Welling (2013), Monte Carlo sampling, as well as stochastic gradient descent (SGD) type algorithms.

Let $\mathcal{F}_{\mathrm{NN},k}^{\mathrm{e}}$ denote the parametric family of encoding probability distributions $P_{U_k|X_k}$ over $\mathcal{U}_k$ for each element on $\mathcal{X}_k$. Each member of this collection, $P_{U_k|X_k;\gamma_k^{\mathrm{e}}}$, is described by a parameter vector $\gamma_k^{\mathrm{e}} \in \Gamma_k^{\mathrm{e}} \subseteq \mathbb{R}^{l_k^{\mathrm{e}}}$, where

$\Gamma_k^{\mathrm{e}} \subseteq \mathbb{R}^{l_k^{\mathrm{e}}}$ denotes the set of allowable parameter vectors. The parameter vector $\boldsymbol{\gamma}_k^{\mathrm{e}}$ is the output of a DNN $f_{\theta_k} : \mathcal{X}_k \to \Gamma_k^{\mathrm{e}}$, with network parameters $\theta_k \in \Theta_k \subseteq \mathbb{R}^{d_k^{\mathrm{e}}}$, e.g., the weights of the network at all layers. The DNN $f_{\theta_k}$ takes $X_k$ as input and outputs the parameter vector $\boldsymbol{\gamma}_k^{\mathrm{e}}$, determining one of the probability members $P_{U_k | X_k ; \gamma_k^{\mathrm{e}}}$. We have

$$\mathcal{F}_{\mathrm{NN},k}^{\mathrm{e}} = \Big\{ P_{U_k | X_k ; \gamma_k^{\mathrm{e}}}(u_k | x_k), \text{ for } u_k \in \mathcal{U}_k, x_k \in \mathcal{X}_k : \quad \boldsymbol{\gamma}_k^{\mathrm{e}} = f_{\theta_k}(x_k), \theta_k \in \Theta_k \Big\}. \qquad (24)$$

For example, the family of multivariate Gaussian distributions is parametrized by the mean $\boldsymbol{\mu}_k^{\theta}$ and covariance matrix $\boldsymbol{\Sigma}_k^{\theta}$, i.e., $\boldsymbol{\gamma}_k := (\boldsymbol{\mu}_k^{\theta}, \boldsymbol{\Sigma}_k^{\theta})$. Therefore, given an observation $X_k$, $\boldsymbol{\gamma}_k := (\boldsymbol{\mu}_k^{\theta}, \boldsymbol{\Sigma}_k^{\theta})$ is determined by the output of the DNN $f_{\theta_k}$ and $\mathcal{F}_{\mathrm{NN},k}^{\mathrm{e}}$ is given by $P_{U_k | X_k ; \gamma_k}(u_k | x_k) = \mathcal{N}(u_k ; \boldsymbol{\mu}_k^{\theta}, \boldsymbol{\Sigma}_k^{\theta})$.

Similarly, for decoders $Q_{Y | U_k}$ over $\mathcal{Y}$, define the family of distributions parametrized by a vector in $\Gamma_k^{\mathrm{d}} \subseteq \mathbb{R}^{l_k^{\mathrm{d}}}$ determined by the output of a DNN $f_{\phi_k} : \mathcal{U}_k \to \Gamma_k^{\mathrm{d}}$, with parameters $\phi_k \in \Phi_k \subseteq \mathbb{R}^{d_k^{\mathrm{d}}}$, as

$$\mathcal{F}_{\mathrm{NN},k}^{\mathrm{d}} = \Big\{ Q_{Y | U_k ; \gamma_k^{\mathrm{d}}}(y | u_k), \text{ for } y \in \mathcal{Y}, u_k \in \mathcal{U}_k : \quad \boldsymbol{\gamma}_k^{\mathrm{d}} = f_{\phi_k}(u_k), \phi_k \in \Phi_k \Big\}, \qquad (25)$$

and for the distribution $Q_{Y | U_\mathcal{K}}$ over $\mathcal{Y}$ for each element in $\mathcal{U}_1 \times \cdots \times \mathcal{U}_K$, define the family of distributions parameterized by the output of the DNN $f_{\phi_\mathcal{K}} : \mathcal{U}_1 \times \cdots \times \mathcal{U}_K \to \Gamma_\mathcal{K}^{\mathrm{d}}$, with $\phi_\mathcal{K} \in \Phi_\mathcal{K} \subseteq \mathbb{R}^{d_\mathcal{K}^{\mathrm{d}}}$, and $\Gamma_\mathcal{K}^{\mathrm{d}} \subseteq \mathbb{R}^{d_\mathcal{K}^{\mathrm{d}}}$

$$\mathcal{F}_{\mathrm{NN},\mathcal{K}}^{\mathrm{d}} = \Big\{ Q_{Y | U_1, \ldots, U_K ; \gamma_\mathcal{K}^{\mathrm{d}}}(y | u_1, \ldots, u_K), y \in \mathcal{Y}, u_k \in \mathcal{U}_k : \gamma_\mathcal{K}^{\mathrm{d}} = f_{\phi_\mathcal{K}}(u_1, \ldots, u_K), \phi_\mathcal{K} \in \Phi_\mathcal{K} \Big\}. \quad (26)$$

Finally, for the distributions $Q_{\varphi_k}(u_k)$ we define the family of distributions with parameter $\boldsymbol{\varphi}_k \in \Psi_k \subseteq \mathbb{R}^{l_k^{\mathrm{p}}}$

$$\mathcal{F}_{\mathrm{NN},k}^{\mathrm{p}} = \Big\{ Q_{U_k ; \varphi_k}(u_k), \text{ for } u_k \in \mathcal{U}_k : \boldsymbol{\varphi}_k \in \Psi_k \Big\}.$$

In the following, for brevity we use $P_{\theta_k}(u_k | x_k)$, $Q_{\psi_k}(y | u_k)$, $Q_{\psi_\mathcal{K}}(y | u_\mathcal{K})$ and $Q_{\varphi_k}(u_k)$ to denote the distributions parametrized by the DNNs $f_{\theta_k}$, $f_{\psi_k}$, $f_{\psi_\mathcal{K}}$ and $\varphi_k$, respectively.

By restricting the optimization of the variational DIB cost in equation 23 to the encoder, decoder and priors within the families of distributions $\mathcal{F}_{\mathrm{NN},k}^{\mathrm{e}}, \mathcal{F}_{\mathrm{NN},k}^{\mathrm{d}}, \mathcal{F}_{\mathrm{NN},\mathcal{K}}^{\mathrm{d}}, \mathcal{F}_{\mathrm{NN},k}^{\mathrm{p}}$ we get

$$\max_{\mathbf{P}} \max_{\mathbf{Q}} \mathcal{L}_s^{\mathrm{VB}}(\mathbf{P}, \mathbf{Q}) \geq \max_{\theta, \phi, \varphi} \mathcal{L}_s^{\mathrm{NN}}(\boldsymbol{\theta}, \boldsymbol{\phi}, \boldsymbol{\varphi}), \qquad (27)$$

where we use the notation $\boldsymbol{\theta} := [\theta_1, \ldots, \theta_K]$, $\boldsymbol{\phi} := [\phi_1, \ldots, \phi_K, \phi_\mathcal{K}]$ and $\boldsymbol{\varphi} := [\varphi_1, \ldots, \varphi_K]$ to denote the DNN and prior parameters and, the cost in equation 27 is given by

$$\mathcal{L}_s^{\mathrm{NN}}(\boldsymbol{\theta}, \boldsymbol{\phi}, \boldsymbol{\varphi}) := \mathbb{E}_{P_{Y,X}} \mathbb{E}_{\{P_{\theta_k}(U_k | X_k)\}} \Big[ \log Q_{\phi_\mathcal{K}}(Y | U_\mathcal{K})$$
$$+ s \sum_{k=1}^{K} \Big( \log Q_{\phi_k}(Y | U_k) - D_{\mathrm{KL}}(P_{\theta_k}(U_k | X_k) \| Q_{\varphi_k}(U_k)) \Big) \Big]. \qquad (28)$$

Next, we train the DNNs to maximize a Monte Carlo approximation of equation 27 over $\boldsymbol{\theta}, \boldsymbol{\phi}, \boldsymbol{\varphi}$ using SGD. We use the reparameterization trick Kingma and Welling (2013), to sample from $P_{\theta_k}(U_k | X_k)$. In particular, we consider $\mathcal{F}_{\mathrm{NN},k}^{\mathrm{e}}$ to consist of a parametric family of distributions that can be sampled by first sampling a random variable $Z_k$ with distribution $P_{Z_k}(z_k)$, $z_k \in \mathcal{Z}_k$ and then transforming the samples using some function $g_{\theta_k} : \mathcal{X}_k \times \mathcal{Z}_k \to \mathcal{U}_k$ parameterized by $\theta_k$, such that $U_k = g_{\theta_k}(x_k, Z_k) \sim P_{\theta_k}(U_k | x_k)$. The reparametrization trick reduces the original optimization to estimating $\theta_k$ of the deterministic function $g_{\theta_k}$ and allows to compute estimates of the gradient using backpropagation Kingma and Welling (2013). The variational DIB cost in equation 27 can be approximated, by sampling $m$ independent samples $\{u_{k,i,j}\}_{j=1}^{m} \sim P_{\theta_k}(u_k | x_{k,i})$ for each training sample $(x_{1,i}, \ldots, x_{K,i}, y_i)$, $i = 1, \ldots, n$. Sampling is performed by using $u_{k,i,j} = g_{\phi_k}(x_{k,i}, z_{k,j})$ with $\{z_{k,j}\}_{j=1}^{m}$ i.i.d. sampled from $P_{Z_k}$. We then have

$$\mathcal{L}_{s,i}^{\mathrm{emp}}(\boldsymbol{\theta}, \boldsymbol{\phi}, \boldsymbol{\varphi}) := \frac{1}{m} \sum_{j=1}^{m} \log Q_{\phi_\mathcal{K}}(y_i | u_{1,i,j}, \ldots, u_{K,i,j})$$
$$+ \frac{s}{m} \sum_{j=1}^{m} \sum_{k=1}^{K} \Big( \log Q_{\phi_k}(y_i | u_{k,i,j}) - D_{\mathrm{KL}}(P_{\theta_k}(U_{k,i} | x_{k,i}) \| Q_{\varphi_k}(U_{k,i})) \Big). \qquad (29)$$

## 4 EXPERIMENTS: RESILIENCE TO NOISE, ROTATION AND OCCLUSION

In this experiment, we test the robustness of our method against noise, rotation and random occlusion on the MNIST dataset. Specifically, we combine two types of random occlusions: the first encoder observes a digit from the MNIST that is occluded by a square which is rotated randomly (rotation angle uniformly distributed over $[-45^o, 45^o]$); and the second encoder observes a noisy version of the same digit corrupted by additive noise

(noise level uniform between 0 and 3). The noisy pixels are clipped between 0 and 1, with more than 60% of the pixels occluded. These occlusions make the problem significantly more involved than the standard MNIST (for which application of our algorithm leads to an relevance of about 99.9%).

We considered a CNN deterministic networks with dropout which achieves a 99.8% for test data on the clean MNIST data. Then, we have trained the same CNN architecture for each of the noisy inputs to the encoders, resulting in a relevance of 92.1% from the input to encoder 1 (randomly rotated occlusion) and 79.68% from the input to encoder 2 (noisy clipped image).

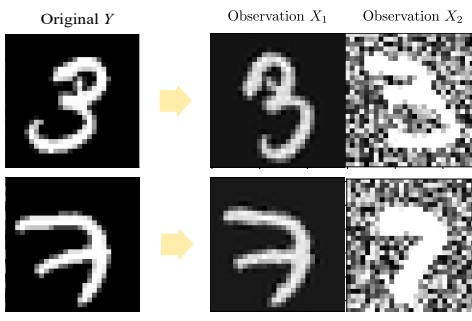

**Figure 3:** View 1: occluded. View 2: noisy.

| | CNN Layers |
|---|---|
| Encoder $k$ | conv. ker. [5,5,32]-ReLu |
| | maxpool [2,2,2] |
| | conv. ker. [5,5,64]-ReLu |
| | maxpool [2,2,2] |
| | dense [1024]-ReLu |
| | dropout 0.4 |
| | dense [256]-relu |
| Latent space $k$ | dense [256]-ReLu |
| Decoder 12 | dense [256]-ReLu |
| Decoder $k$ | dense [256]-ReLu |

**Table 1:** Used CNN architecture.

**Figure 4:** relevance v.s. sum-complexity for $n = 50.000$ and $s \in [10^{-10}, 1]$.

| | relevance (%) | |
|---|---|---|
| | 1 shot | avg. |
| D-VIB | **96.16** | **97.24** |
| D-VIB-noReg | 96.04 | 96.72 |
| C-VIB | 96.01 | 96.68 |
| Deterministic CNN | 93.18 | 93.18 |
| Independent CNNs | 92.1 / 79.68 | 93.1 / 82.01 |

**Table 2:** Achieved relevance levels.

We applied our D-VIB algorithm of Section 3.4 to this model with the CNN architecture of Table 1, in which Encoder $k = 1, 2$ is parametrized by an $n_{u_k} = 256$ dimensional multivariate Gaussian distribution $\mathcal{N}(\boldsymbol{\mu}_k^{\mathrm{e}}, \boldsymbol{\Sigma}_k^{\mathrm{e}})$ determined by the output of a DNN $f_{\theta_k}$ consisting of the concatenation of convolution, dense and maxpool layers with ReLu activations and dropout. The output of the last layer is followed by a dense layer without activation that generate $\boldsymbol{\mu}_k^{\mathrm{e}}$ and $\boldsymbol{\Sigma}_k^{\mathrm{e}}$. The prior is chosen as $Q_{\varphi_k}(\mathbf{u}) = \mathcal{N}(\mathbf{0}, \mathbf{I})$. Each decoder takes the samples from $P_{\theta_k}(U_k|X_k)$ and processes its inputs with a dense layer DNN ($f_{\phi_{\mathcal{K}}}$ and $f_{\phi_k}$) each with 256 neurons and ReLu activation, which outputs a vector $\hat{\mathbf{y}}_i$ of size $|\mathcal{Y}| = 10$ normalized with a softmax, corresponding to a distribution over the one-hot encoding of the digit labels $\{0, \dots, 9\}$ from the $K$ observations,

$$Q_{\phi_k}(\hat{\mathbf{y}}_k|\mathbf{u}_k) = \mathrm{Softmax}(f_{\phi_k}(U_k)), \quad k = 1, 2, \text{ and} \qquad (30)$$

$$Q_{\phi_{\mathcal{K}}}(\hat{\mathbf{y}}|\mathbf{u}_{\mathcal{K}}) = \mathrm{Softmax}(f_{\phi_{\mathcal{K}}}(U_1, U_2)), \qquad (31)$$

where $\mathrm{Softmax}(\mathbf{p})$ for $\mathbf{p} \in \mathbb{R}^d$ is a vector with $i$-th entry as $[\mathrm{Softmax}(\mathbf{p})]_i = \exp(p_i) / \sum_{j=1}^d \exp(p_j)$. Figure 4 shows the relevance-complexity tradeoffs obtained using our D-VIB algorithm of Section 3.4, with $n = 50.000$ and 15 distinct $s$-values randomly chosen in the range $[10^{-10}, 1]$. For comparison, we also present the performance obtained using three methods among state-of the-art multiview learning approaches: (i) applying a deterministic CNN on the two views concatenated (deterministic CNN), (ii) applying the single-encoder variational IB method of Alemi et al. on the two views concatenated (C-VIB), and (iii) learning one function for each view via a distinct CNNs and optimize all CNNs independently (independent CNNs). The achieved relevance is reported in Table 2. For other experimental results, see the appendices section.

We also mention that at a high level our algorithm D-VIB can be considered as performing some form of co-regularization (for instance its Gaussian version is similar to the CCA of Hardoon et al. (2004)). Comparatively, the single-view algorithm C-VIB can be viewed as belonging to the family of co-training style algorithms (such as the co-EM of Nigam and Ghani (2000)) which, as mentioned in the recent survey Zhao et al. (2017), override on single-view algorithms. The performance of D-VIB dominates that of C-VIB, which itself dominates co-EM.

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

# APPENDICES

# PROOFS OF MAIN THEOREMS, PROPOSITIONS AND LEMMAS. ADDITIONAL EXPERIMENTAL RESULTS

## 5 PROOFS OF MAIN THEOREMS, PROPOSITIONS AND LEMMAS

### 5.1 AUXILIARY LEMMAS

**Lemma 2** *Dembo et al. (1991); Ekrem and Ulukus (2014) Let $(\mathbf{X}, \mathbf{Y})$ be a pair of random vectors with pmf $p(\mathbf{x}, \mathbf{y})$. We have*

$$\log |(\pi e)\mathbf{J}^{-1}(\mathbf{X}|\mathbf{Y})| \leq h(\mathbf{X}|\mathbf{Y}) \leq \log |(\pi e)\mathrm{mmse}(\mathbf{X}|\mathbf{Y})|,$$

*where the conditional Fischer information matrix is defined as*

$$\mathbf{J}(\mathbf{X}|\mathbf{Y}) := \mathrm{E}[\nabla \log p(\mathbf{X}|\mathbf{Y})\nabla \log p(\mathbf{X}|\mathbf{Y})^{\dagger}],$$

*and the minimum mean squared error (MMSE) matrix is*

$$\mathrm{mmse}(\mathbf{X}|\mathbf{Y}) := \mathrm{E}[(\mathbf{X} - \mathrm{E}[\mathbf{X}|\mathbf{Y}])(\mathbf{X} - \mathrm{E}[\mathbf{X}|\mathbf{Y}])^{\dagger}].$$

**Lemma 3** *Ekrem and Ulukus (2014) Let $(\mathbf{V}_1, \mathbf{V}_2)$ be a random vector with finite second moments and $\mathbf{N} \sim \mathcal{CN}(\mathbf{0}, \mathbf{\Sigma}_N)$ independent of $(\mathbf{V}_1, \mathbf{V}_2)$. Then*

$$\mathrm{mmse}(\mathbf{V}_2|\mathbf{V}_1, \mathbf{V}_2 + \mathbf{N}) = \mathbf{\Sigma}_N - \mathbf{\Sigma}_N \mathbf{J}(\mathbf{V}_2 + \mathbf{N}|\mathbf{V}_1)\mathbf{\Sigma}_N.$$

### 5.2 PROOF OF THEOREM 1

If $K = 1$ the distributed learning problem that we study boils down to the well known Information Bottleneck (IB) problem of Tishby et al. (1999). The single-encoder IB problem is essentially a remote point-to-point source coding problem Dobrushin and Tsybakov (1962) in which distortion is measured under the logarithm loss fidelity criterion Harremoes and Tishby (2007). In accordance with this analogy, for $K \geq 2$ consider the multiterminal source coding problem under logarithmic loss in which the sequence $Y^n$ models a remote source that is observed by $K$ spatially distributed agents; the agents observe noisy versions of the remote source and communicate independently with a decoder or Chief Executive Officer (CEO) over rate-constrained noise-free links. For instance, agent $k$, $k \in \mathcal{K}$, observes $X_k^n$ and uses $R_k$ bits per sample to describe it to the decoder. The decoder wants to reconstruct the remote source $Y^n$ to within a prescribed fidelity level, where incurred distortion is measured using the logarithmic loss criterion, i.e.,

$$\ell_{\log}(y^n, \hat{y}^n) = \frac{1}{n} \log \frac{1}{\hat{P}_{Y^n|J}(y^n|\phi_1(x_1^n), \ldots, \phi_K(x_K^n))}, \tag{32}$$

where $J = (\phi_1(X_1^n), \ldots, \phi_K(X_K^n))$.

Here, $(X_1^n, \ldots, X_K^n, Y^n)$ is assumed to be distributed i.i.d. according to the $n$-product of the pmf $P_{X_1, \ldots, X_K, Y}$, i.e., the Markov chain equation 3 holds.

**Definition 2** *A rate-distortion code (of blocklength $n$) for the CEO problem consists of $K$ encoding functions*

$$\tilde{\phi}_k : \mathcal{X}_k^n \to \{1, \ldots, M_k^{(n)}\}, \quad \text{for } k = 1, \ldots, K, \tag{33}$$

*and a decoding function*

$$\tilde{\psi} \; : \; \{1, \ldots, M_1^{(n)}\} \times \ldots \times \{1, \ldots, M_K^{(n)}\} \to \hat{\mathcal{Y}}^n . \square \tag{34}$$

*A distortion-rate tuple $(D, R_1, \ldots, R_K)$ is achievable for the DM CEO source coding problem with side information if there exist a blocklength $n$, encoding functions $\{\tilde{\phi}_k\}_{k=1}^K$ and a decoding function $\tilde{\psi}$ such that*

$$R_k \geq \frac{1}{n} \log M_k^{(n)}, \quad for \; k = 1, \ldots, K,$$
$$D \geq \mathbb{E}\big[\ell_{\log}\big(Y^n, \tilde{\psi}(\tilde{\phi}_1(X_1^n), \ldots, \tilde{\phi}_K(X_K^n))\big)\big].$$

*The distortion-rate region $\mathcal{DR}_{\mathrm{CEO}}$ of the CEO model is defined as the closure of all non-negative tuples $(D, R_1, \ldots, R_K)$ that are achievable.* □

Key to the proof of Theorem 1 is the following proposition which states that $\mathcal{IR}_{\mathrm{DIB}}$ and $\mathcal{DR}_{\mathrm{CEO}}$ can be inferred from each other.

**Proposition 3** $(\Delta, R_1, \ldots, R_K) \in \mathcal{IR}_{\mathrm{DIB}}$ *if and only if* $\big(H(Y) - \Delta, R_1, \ldots, R_K\big) \in \mathcal{DR}_{\mathrm{CEO}}$.

**Proof:** Let, for $k = 1, \ldots, K$, $J_k = \phi_k(X_k^n)$ and $J = (J_1, \ldots, J_K)$. Then,

$$\mathbb{E}[\ell_{\log}(Y^n, \hat{Y}^n)|J = j] = \sum_{y^n \in \mathcal{Y}^n} P(y^n|j) \log \left( \frac{1}{\hat{P}(y^n|j)} \right) \tag{35}$$

$$= \sum_{y^n \in \mathcal{Y}^n} P(y^n|j) \log \left( \frac{P(y^n|j)}{\hat{P}(y^n|j)} \right) + H(Y^n|J = j) \tag{36}$$

$$= D_{\mathrm{KL}}(P(y^n|j)\|\hat{P}(y^n|j)) + H(Y^n|J = j) \tag{37}$$

$$\geq H(Y^n|J = j), \tag{38}$$

where equation 38 is due to the non-negativity of the Kullback-Leibler divergence and the equality holds if and only if for $\hat{P}(y^n|j) = P(y^n|j)$ where $P(y^n|j) = \Pr\{Y^n = y^n|J = j\}$ for all $j$ and $y^n \in \mathcal{Y}^n$.

Let an achievable tuple $(\Delta, R_1, \ldots, R_K) \in \mathcal{IR}_{\mathrm{DIB}}$ be given. Then, there must exist functions $\{\phi_k\}_{k=1}^K$ such that equation 9 and equation 10 hold. Using equation 38 that by letting the decoding function $\tilde{\psi}(J_\mathcal{K}) = \{P_{Y^n|J_\mathcal{K}}(y^n|J_\mathcal{K})\}$, we have $\mathbb{E}[\ell_{\log}(Y^n, \hat{Y}^n)|J_\mathcal{K}] = H(Y^n|J_\mathcal{K})$, which implies $(H(Y) - \Delta, R_1, \ldots, R_K) \in \mathcal{DR}_{\mathrm{CEO}}$.

The result of Theorem 1 follows easily by combining (Courtade and Weissman, 2014, Theorem 10), which provides a single-letter characterization of the rate distortion region $\mathcal{DR}_{\mathrm{CEO}}^\star$ of the CEO problem, and Proposition 3.

## 5.3 PROOF OF THEOREM 2

The proof of the direct part of Theorem 2 follows by evaluating the region of Theorem 1 with the choice $T = \emptyset$ and $p(\mathbf{u}_k|\mathbf{x}_k, t) = \mathcal{CN}(\mathbf{x}_k, \boldsymbol{\Sigma}_k^{1/2}(\boldsymbol{\Omega}_k - \mathbf{I})\boldsymbol{\Sigma}_k^{1/2})$.

The proof of the converse part is as follows. Fix $t \in \mathcal{T}$, $\mathcal{S} \subseteq \mathcal{K}$ and a family of distributions $\{p(\mathbf{u}_k|\mathbf{x}_k, t)\}_{k=1}^K$ such that the joint distribution factorizes as equation 13. Also, let $\mathbf{0} \preceq \boldsymbol{\Omega}_{k,t} \preceq \boldsymbol{\Sigma}_k^{-1}$ and

$$\mathrm{mmse}(\mathbf{X}_k|\mathbf{Y}, \mathbf{U}_{k,t}, t) = \boldsymbol{\Sigma}_k - \boldsymbol{\Sigma}_k\boldsymbol{\Omega}_{k,t}\boldsymbol{\Sigma}_k. \tag{39}$$

Such $\boldsymbol{\Omega}_{k,t}$ always exists since

$$\mathbf{0} \preceq \mathrm{mmse}(\mathbf{X}_k|\mathbf{Y}, \mathbf{U}_{k,t}, t) \preceq \boldsymbol{\Sigma}_k^{-1}. \tag{40}$$

Then, we have

$$I(\mathbf{X}_k; \mathbf{U}_k|\mathbf{Y}, t) \geq \log |\boldsymbol{\Sigma}_k| - \log |\mathrm{mmse}(\mathbf{X}_k|\mathbf{Y}, \mathbf{U}_{k,t}, t)|$$
$$= -\log |\mathbf{I} - \boldsymbol{\Sigma}_k^{1/2}\boldsymbol{\Omega}_{k,t}\boldsymbol{\Sigma}_k^{1/2}|, \tag{41}$$

where the inequality is due to Lemma 2; and equation 41 is due to equation 39.

Also, we have

$$I(\mathbf{Y}; \mathbf{U}_{S^c,t}|t) \leq \log |\boldsymbol{\Sigma}_\mathbf{y}| - \log |\mathbf{J}^{-1}(\mathbf{Y}|\mathbf{U}_{S^c,t}, t)| \tag{42}$$

$$= \log \left| \sum_{k \in \mathcal{S}^c} \boldsymbol{\Sigma}_{\mathbf{y}}^{1/2} \mathbf{H}_k^\dagger \boldsymbol{\Omega}_{k,t} \mathbf{H}_k \boldsymbol{\Sigma}_{\mathbf{y}}^{1/2} + \mathbf{I} \right|, \tag{43}$$

where equation 42 follows by using Lemma 2; and equation 43 holds by using the following equality

$$\mathbf{J}(\mathbf{Y}|\mathbf{U}_{\mathcal{S}^c,t}, t) = \sum_{k \in \mathcal{S}^c} \mathbf{H}_k^\dagger \boldsymbol{\Omega}_{k,t} \mathbf{H}_k + \boldsymbol{\Sigma}_{\mathbf{y}}^{-1}. \tag{44}$$

the proof of which uses a connection between MMSE and Fisher information as shown next.

For the proof of equation 44, first note that from the MMSE estimation of Gaussian random vectors El Gamal and Kim (2011), we have

$$\mathbf{Y} = \mathrm{E}[\mathbf{Y}|\mathbf{X}_{\mathcal{S}^c}] + \mathbf{Z}_{\mathcal{S}^c} = \sum_{k \in \mathcal{S}^c} \mathbf{G}_k \mathbf{X}_k + \mathbf{Z}_{\mathcal{S}^c}, \tag{45}$$

where $\mathbf{G}_k = \boldsymbol{\Sigma}_{\mathbf{y}|\mathbf{x}_{\mathcal{S}^c}} \mathbf{H}_k^\dagger \boldsymbol{\Sigma}_k^{-1}$ and $\mathbf{Z}_{\mathcal{S}^c} \sim \mathcal{CN}(\mathbf{0}, \boldsymbol{\Sigma}_{\mathbf{y}|\mathbf{x}_{\mathcal{S}^c}})$, with

$$\boldsymbol{\Sigma}_{\mathbf{y}|\mathbf{x}_{\mathcal{S}^c}}^{-1} = \boldsymbol{\Sigma}_{\mathbf{y}}^{-1} + \sum_{k \in \mathcal{S}^c} \mathbf{H}_k^\dagger \boldsymbol{\Sigma}_k^{-1} \mathbf{H}_k. \tag{46}$$

Note that $\mathbf{Z}_{\mathcal{S}^c}$ is independent of $\mathbf{Y}_{\mathcal{S}^c}$ due to the orthogonality principle of the MMSE and its Gaussian distribution. Hence, it is also independent of $\mathbf{U}_{\mathcal{S}^c, q}$. We have

$$\mathrm{mmse}\left( \sum_{k \in \mathcal{S}^c} \mathbf{G}_k \mathbf{X}_k \Big| \mathbf{Y}, \mathbf{U}_{\mathcal{S}^c, t}, t \right) = \sum_{k \in \mathcal{S}^c} \mathbf{G}_k \mathrm{mmse}\left( \mathbf{X}_k | \mathbf{Y}, \mathbf{U}_{\mathcal{S}^c, t}, t \right) \mathbf{G}_k^\dagger \tag{47}$$

$$= \boldsymbol{\Sigma}_{\mathbf{y}|\mathbf{x}_{\mathcal{S}^c}} \sum_{k \in \mathcal{S}^c} \mathbf{H}_k^\dagger \left( \boldsymbol{\Sigma}_k^{-1} - \boldsymbol{\Omega}_k \right) \mathbf{H}_k \boldsymbol{\Sigma}_{\mathbf{y}|\mathbf{x}_{\mathcal{S}^c}}, \tag{48}$$

where equation 47 follows since the cross terms are zero due to the Markov chain $(\mathbf{U}_{k,t}, \mathbf{X}_k) \multimap \mathbf{Y} \multimap (\mathbf{U}_{\mathcal{K}/k,t}, \mathbf{X}_{\mathcal{K}/k})$; and equation 48 follows due to equation 39 and $\mathbf{G}_k$. Finally,

$$\mathbf{J}(\mathbf{Y}|\mathbf{U}_{\mathcal{S}^c,t}, t) = \boldsymbol{\Sigma}_{\mathbf{y}|\mathbf{x}_{\mathcal{S}^c}}^{-1} - \boldsymbol{\Sigma}_{\mathbf{y}|\mathbf{x}_{\mathcal{S}^c}}^{-1} \mathrm{mmse}\left( \sum_{k \in \mathcal{S}^c} \mathbf{G}_k \mathbf{X}_k \Big| \mathbf{Y}, \mathbf{U}_{\mathcal{S}^c,t}, t \right) \boldsymbol{\Sigma}_{\mathbf{y}|\mathbf{x}_{\mathcal{S}^c}}^{-1} \tag{49}$$

$$= \boldsymbol{\Sigma}_{\mathbf{y}|\mathbf{x}_{\mathcal{S}^c}}^{-1} - \sum_{k \in \mathcal{S}^c} \mathbf{H}_k^\dagger \left( \boldsymbol{\Sigma}_k^{-1} - \boldsymbol{\Omega}_{k,t} \right) \mathbf{H}_k \tag{50}$$

$$= \boldsymbol{\Sigma}_{\mathbf{y}}^{-1} + \sum_{k \in \mathcal{S}^c} \mathbf{H}_k^\dagger \boldsymbol{\Omega}_{k,t} \mathbf{H}_k, \tag{51}$$

where equation 49 is due to Lemma 3; equation 50 is due to equation 48; and equation 51 follows due to equation 46.

Now, let $\bar{\boldsymbol{\Omega}}_k := \sum_{t \in \mathcal{T}} p(t) \boldsymbol{\Omega}_{k,t}$. The rest of the converse proof follows by averaging over the time sharing random variable to get

$$I(\mathbf{X}_k; \mathbf{U}_k | \mathbf{Y}, T) \geq - \sum_{t \in \mathcal{T}} p(t) \log |\mathbf{I} - \boldsymbol{\Sigma}_k^{1/2} \boldsymbol{\Omega}_{k,t} \boldsymbol{\Sigma}_k^{1/2}|$$

$$\geq - \log |\mathbf{I} - \boldsymbol{\Sigma}_k^{1/2} \bar{\boldsymbol{\Omega}}_k \boldsymbol{\Sigma}_k^{1/2}|, \tag{52}$$

where equation 52 follows from the concavity of the log-det function and Jensen's inequality. Similarly to equation 52, from equation 43 and Jensen's Inequality we have

$$I(\mathbf{Y}; \mathbf{U}_{\mathcal{S}^c} | T) \leq \log \left| \sum_{k \in \mathcal{S}^c} \boldsymbol{\Sigma}_{\mathbf{y}}^{1/2} \mathbf{H}_k^\dagger \bar{\boldsymbol{\Omega}}_k \mathbf{H}_k \boldsymbol{\Sigma}_{\mathbf{y}}^{1/2} + \mathbf{I} \right|. \tag{53}$$

Finally, using equation 52 and equation 53 in equation **??**, noting that $\boldsymbol{\Omega}_k = \sum_{t \in \mathcal{T}} p(t) \boldsymbol{\Omega}_{k,t} \preceq \boldsymbol{\Sigma}_k^{-1}$ since $\mathbf{0} \preceq \boldsymbol{\Omega}_{k,t} \preceq \boldsymbol{\Sigma}_k^{-1}$, and taking the union over $\boldsymbol{\Omega}_k$ satisfying $\mathbf{0} \preceq \boldsymbol{\Omega}_k \preceq \boldsymbol{\Sigma}_k^{-1}$, completes the proof of the converse part; and, hence, that of Theorem 2.

### 5.4 PROOF OF PROPOSITION 1

For simplicity of exposition, the proof is given for the case $K = 2$ encoders. The proof for $K > 2$ follows similarly. By the definition of $\mathcal{IR}_{\text{DIB}}^{\text{sum}}$, the accuracy complexity tuple $(\Delta, R_{\text{sum}}) \in \mathbb{R}_+^2$ is achievable for some random variables $Y, X_1, X_2, U_1, U_2$ with joint pmf satisfying equation 13, if it holds that

$$\Delta \leq I(Y; U_1, U_2) \tag{54}$$
$$\Delta \leq R_1 - I(X_1; U_1|Y) + I(Y; U_2) \tag{55}$$
$$\Delta \leq R_2 - I(X_2; U_2|Y) + I(Y; U_1) \tag{56}$$
$$\Delta \leq R_1 + R_2 - I(X_1; U_1|Y) - I(X_2; U_2|Y) \tag{57}$$
$$R_1 + R_2 \leq R_{\text{sum}}. \tag{58}$$

The application of the Fourier-Motzkin elimination to project out $R_1$ and $R_2$ reduces the system on inequalities equation 54-equation 58 to the following system of inequalities

$$\Delta \leq I(Y; U_1, U_2) \tag{59}$$
$$\Delta \leq R_{\text{sum}} - I(X_1; U_1|Y) - I(X_2; U_2|Y) \tag{60}$$
$$2\Delta \leq R_{\text{sum}} - I(X_1; U_1|Y) - I(X_2; U_2|Y) + I(Y; U_1) + I(Y; U_2) \tag{61}$$

It follows due to the Markov chain $U_1 \multimap X_1 \multimap Y \multimap X_2 \multimap U_2$ that we have $I(Y; U_1, U_2) \leq I(Y; U_1) + I(Y; U_2)$. Therefore, inequality equation 61 is redundant as it is implied by equation 59 and equation 60. This completes the proof of Proposition 1.

### 5.5 PROOF OF PROPOSITION 2

Suppose that $\mathbf{P}^*$ yields the maximum in equation 16. Then,

$$(1 + s)\Delta_s = (1 + sK)H(Y) + sR_s + \mathcal{L}_s(\mathbf{P}^*) \tag{62}$$

$$= (1 + sK)H(Y) + sR_s + \left( -H(Y|U_{\mathcal{K}}^*) - s\sum_{k=1}^{K}[H(Y|U_k^*) + I(X_k; U_k^*)] \right) \tag{63}$$

$$= (1 + sK)H(Y) + sR_s + (-H(Y|U_{\mathcal{K}}^*) - s(R_s - I(Y; U_{\mathcal{K}}^*) + KH(Y))) \tag{64}$$
$$= (1 + s)I(Y; U_{\mathcal{K}}^*) \tag{65}$$
$$\leq (1 + s)\Delta(R_s, P_{X_{\mathcal{K}}, Y}), \tag{66}$$

where equation 63 is due to the definition of $\mathcal{L}_s(\mathbf{P})$ in equation 18; equation 64 follows since we have $\sum_{k=1}^{K}[I(X_k; U_k^*) + H(Y|U_k^*)] = R_s - I(Y; U_{\mathcal{K}}^*) + KH(Y)$ from the definition of $R_s$ in equation 17; and equation 66 follows from the definition in equation ??.

Conversely, if $\mathbf{P}^*$ is the solution to the maximization in the function $\Delta(R_{\text{sum}}, P_{X_{\mathcal{K}}, Y})$ in equation ?? such that $\Delta(R_{\text{sum}}, P_{X_{\mathcal{K}}, Y}) = \Delta_s$, then $\Delta_s \leq I(Y; U_{\mathcal{K}}^*)$ and $\Delta_s \leq R_{\text{sum}} - \sum_{k=1}^{K} I(X_k; U_k^*|Y)$ and we have, for any $s \geq 0$, that

$$\Delta(R_{\text{sum}}, P_{X_{\mathcal{K}}, Y}) = \Delta_s$$

$$\leq \Delta_s - (\Delta_s - I(Y; U_{\mathcal{K}}^*)) - s\left( \Delta_s - R_{\text{sum}} + \sum_{k=1}^{K} I(X_k; U_k^*|Y) \right)$$

$$= I(Y; U_{\mathcal{K}}^*) - s\Delta_s + sR_{\text{sum}} - s\sum_{k=1}^{K} I(X_k; U_k^*|Y)$$

$$= H(Y) - s\Delta_s + sR_{\text{sum}} - H(Y|U_{\mathcal{K}}^*) - s\sum_{k=1}^{K}[I(X_k; U_k^*) + H(Y|U_k^*)] + sKH(Y) \tag{67}$$

$$\leq H(Y) - s\Delta_s + sR_{\text{sum}} + \mathcal{L}_s^* + sKH(Y) \tag{68}$$
$$= H(Y) - s\Delta_s + sR_{\text{sum}} + sKH(Y) - ((1 + sK)H(Y) + sR_s - (1 + s)\Delta_s) \tag{69}$$
$$= \Delta_s + s(R_{\text{sum}} - R_s), \tag{70}$$

where in equation 67 we have $\sum_{k=1}^{K} I(X_k; U_k|Y) = -KH(Y) + \sum_{k=1}^{K} I(X_k; U_k) + H(Y|U_k)$ due to the Markov chain $U_k - X_k - Y - (X_{\mathcal{K}\backslash k}, U_{\mathcal{K}\backslash k})$; equation 68 follows since $\mathcal{L}_s^*$ is the maximum over all possible distributions $\mathbf{P}$ (not necessarily $\mathbf{P}^*$ maximizing $\Delta(R_{\text{sum}}, P_{X_{\mathcal{K}}, Y})$); and equation 69 is due to equation 16.

Finally, equation 70 is valid for any $R_{\text{sum}} \geq 0$ and $s \geq 0$. Given $s$, and hence $(\Delta_s, R_s)$, choosing $R = R_s$ yields $\Delta(R_s, P_{X_{\mathcal{K}}, Y}) \leq \Delta_s$. Together with equation 66, this completes the proof of Proposition 2.

## 5.6 PROOF OF LEMMA 1

The proof follows by deriving the following bounds. For any conditional pmf $Q_{Y|Z}(y|z)$, $y \in \mathcal{Y}$ and $z \in \mathcal{Z}$, e.g., $\mathcal{Z} = \mathcal{U}_{\mathcal{K}}$ or $\mathcal{Z} = \mathcal{U}_k$, proceeding similarly to equation 38 and averaging over $Z$, we have

$$H(Y|Z) = \mathbb{E}[-\log Q_{Y|Z}(Y|Z)] - D_{\text{KL}}(P_{Y|Z} \| Q_{Y|Z}). \tag{71}$$

Similarly, we have

$$I(X_k; U_k) = H(U_k) - H(U_k|X_k) \tag{72}$$

$$= \mathbb{E}[-\log Q_{U_k}(U_k)] - D_{\text{KL}}(P_{U_k} \| Q_{U_k}) - H(X_k|U_K) \tag{73}$$

$$= D_{\text{KL}}(P_{Y|U_k} \| Q_{U_k}) - D_{\text{KL}}(P_{U_k} \| Q_{U_k}) \tag{74}$$

Thus, we get

$$\mathcal{L}_s(\mathbf{P}) = \mathcal{L}_s^{\text{VB}}(\mathbf{P}, \mathbf{Q}) + D_{\text{KL}}(P_{Y|U_{\mathcal{K}}} \| Q_{Y|U_{\mathcal{K}}}) + s \sum_{k=1}^{K} (D_{\text{KL}}(P_{Y|U_k} \| Q_{Y|U_k}) + D_{\text{KL}}(P_{U_k} \| Q_{U_k}))$$

$$\geq \mathcal{L}_s^{\text{VB}}(\mathbf{P}, \mathbf{Q}), \tag{75}$$

where equation 75 holds by the non-negativity of relative entropy: and the equality is met if and only if $\mathbf{Q}^*$ is as given by equation 21 and equation 22.

# 6 OTHER EXPERIMENTAL RESULTS (REGRESSION FOR UNKNOWN GAUSSIAN MODEL)

## 6.1 D-VIB ALGORITHM FOR VECTOR GAUSSIAN MODEL

For the vector Gaussian data model equation 14 the optimal distributions $\mathbf{P}$ and $\mathbf{Q}$ in equation 23 lie within the family of multivariate Gaussian distributions. Motivated by this observation, we consider the following parameterization for $k \in \mathcal{K}$:

$$P_{\theta_k}(\mathbf{u}_k|\mathbf{x}_k) = \mathcal{N}(\mathbf{u}_k; \boldsymbol{\mu}_k^{\text{e}}, \boldsymbol{\Sigma}_k^{\text{e}}) \tag{76}$$

$$Q_{\phi_{\mathcal{K}}}(\hat{\mathbf{y}}|\mathbf{u}_{\mathcal{K}}) = \mathcal{N}(\hat{\mathbf{y}}; \boldsymbol{\mu}_{\mathcal{K}}^{\text{d}}, \boldsymbol{\Sigma}_{\mathcal{K}}^{\text{d}}) \tag{77}$$

$$Q_{\phi_k}(\hat{\mathbf{y}}|\mathbf{u}_k) = \mathcal{N}(\hat{\mathbf{y}}; \boldsymbol{\mu}_k^{\text{d}}, \boldsymbol{\Sigma}_k^{\text{d}}) \tag{78}$$

$$Q_{\varphi_k}(\mathbf{u}_k) = \mathcal{N}(\mathbf{0}, \mathbf{I}). \tag{79}$$

where $\boldsymbol{\mu}_k^{\text{e}}, \boldsymbol{\Sigma}_k^{\text{e}}$ are the output of a DNN $f_{\theta_k}$ with input $\mathbf{X}_k$ that encodes the observations in a $n_{u_k}$-dimensional Gaussian distribution, $\boldsymbol{\mu}_{\mathcal{K}}^{\text{d}}, \boldsymbol{\Sigma}_{\mathcal{K}}^{\text{d}}$ are the outputs of a DNN $f_{\phi_{\mathcal{K}}}$ with inputs $\mathbf{U}_1, \dots, \mathbf{U}_K$, sampled from $P_{\theta_k}(\mathbf{u}_k|\mathbf{x}_k)$, and $\boldsymbol{\mu}_k^{\text{d}}, \boldsymbol{\Sigma}_k^{\text{e}}$ are the output of a DNN $f_{\phi_k}$ with input $\mathbf{U}_k$, $k = 1, \dots, K$.

With the above choice of parametric encoders and decoders, and using a single sample $m = 1$, the empirical DIB cost in equation 29 is given for the sample $(\mathbf{x}_{1,i}, \dots, \mathbf{x}_{K,i}, \mathbf{y}_i)$ by

$$\mathcal{L}_{s,i}^{\text{emp}}(\boldsymbol{\theta}, \boldsymbol{\phi}, \boldsymbol{\varphi}) := -\frac{1}{2} \left( (\boldsymbol{y}_i - \boldsymbol{\mu}_{12,i}^{\text{d}})^T \boldsymbol{\Sigma}_{12,i}^{\text{d},-1} (\boldsymbol{y}_i - \boldsymbol{\mu}_{12,i}^{\text{d}}) + \log \det(\boldsymbol{\Sigma}_{12,i}^{\text{d}}) \right)$$

$$- s \sum_{k=1}^{K} \frac{1}{2} \left( (\boldsymbol{y}_i - \boldsymbol{\mu}_{k,i}^{\text{d}})^T \boldsymbol{\Sigma}_{k,i}^{\text{d}-1} (\boldsymbol{y}_i - \boldsymbol{\mu}_{k,i}^{\text{d}}) + \log \det(\boldsymbol{\Sigma}_{k,i}^{\text{d}}) \right)$$

$$- s \sum_{k=1}^{K} \frac{1}{2} \left( (\boldsymbol{\mu}_{k,i}^{\text{e}} - \mathbf{I})^T (\boldsymbol{\mu}_{k,i}^{\text{e}} - \mathbf{I}) + \log |\boldsymbol{\Sigma}_{k,i}^{\text{e},-1}| - n_{u_k} + \text{tr}\{\boldsymbol{\Sigma}_{k,i}^{\text{e}}\} \right)$$

$$- \frac{n_y}{2}(1 + sK) \log(2\pi),$$

where $(\boldsymbol{\mu}_{12,i}^{\text{d}}, \boldsymbol{\Sigma}_{12,i}^{\text{d}})$ denote the output of the DNN $f_{\phi_{\mathcal{K}}}$ for the $i$-th sample $(\mathbf{x}_{1,i}, \dots, \mathbf{x}_{K,i}, \mathbf{y}_i)$, and similarly for the other mean and covariance terms; and where we have used that each term in the empirical DIB cost equation 29 can be computed noting that for $d$-dimensional Gaussian pmfs $\mathcal{N}(\mathbf{y}; \boldsymbol{\mu}, \boldsymbol{\Sigma})$ we have

$$\log \mathcal{N}(\mathbf{y}; \boldsymbol{\mu}, \boldsymbol{\Sigma}) = -\frac{1}{2} \left( (\boldsymbol{y} - \boldsymbol{\mu})^T \boldsymbol{\Sigma}^{-1} (\boldsymbol{y} - \boldsymbol{\mu}) + d \log(2\pi) + \log \det(\boldsymbol{\Sigma}) \right),$$

and the KL divergence between two multivariate Gaussian pmfs $P_1 \sim \mathcal{N}(\boldsymbol{\mu}_1, \boldsymbol{\Sigma}_1)$ and $P_2 \sim \mathcal{N}(\boldsymbol{\mu}_2, \boldsymbol{\Sigma}_2)$ in $\mathbb{R}^d$, is

$$D_{\mathrm{KL}}(P_1 \| P_2) = \frac{1}{2} \left( (\boldsymbol{\mu}_1 - \boldsymbol{\mu}_2)^T \boldsymbol{\Sigma}_2^{-1} (\boldsymbol{\mu}_1 - \boldsymbol{\mu}_2) + \log |\boldsymbol{\Sigma}_2 \boldsymbol{\Sigma}_1^{-1}| - d + \mathrm{tr}\{\boldsymbol{\Sigma}_2^{-1} \boldsymbol{\Sigma}_1\} \right). \tag{80}$$

The multivariate Gaussian parametrization of the encoders, decoders and prior distribution as given by equation 76-equation 79 can be used for other data models that are not necessary Gaussian. For example, it is particularly suitable for regression problems in which $Y$ lies on a continuous space. Also, it is very often used in conjunction with VAE generative problems Rezende et al. (2014); Kingma and Welling (2013).

## 6.2 Regression for Vector Gaussian Data Model

Consider a distributed learning model with $K = 2$ encoders, each observing a noisy version of an $n_y$-dimensional Gaussian vector $\mathbf{Y} \sim \mathcal{N}(\mathbf{y}; \mathbf{0}, \mathbf{I})$, as $\mathbf{X}_k = \mathbf{H}_k \mathbf{Y} + \mathbf{N}_k$, where $\mathbf{H}_k \in \mathbb{R}^{n_k \times n_y}$ and the noises are distributed as $\mathbf{N}_k \sim \mathcal{N}(\mathbf{0}, \mathbf{I})$ for $k = 1, 2$.

For this model, the optimal accuracy-complexity region can be computed using Theorem 2. In what follows, we evaluate the performance of our D-VIB of the previous section for regression. The algorithm is trained using a dataset of $n$ i.i.d. samples $\{(\mathbf{X}_{1,i}, \mathbf{X}_{2,i}, \mathbf{Y}_i)\}_{i=1}^{n}$ form the described vector Gaussian data model. We train the DNNs for various values of the parameter $s$. We use the multivariate Gaussian parameterization in equation 76-equation 79 for the DNNs architecture shown in Table 6.2. Specifically, Encoder $k$, $k = 1, 2$, consists of three dense layers of 512 neurons each followed by rectified linear unit (ReLu) activations. The output of encoder $k$ is processed by a dense layer without nonlinear activation to generate $\boldsymbol{\mu}_k^{\mathrm{e}}$ and $\boldsymbol{\Sigma}_k^{\mathrm{e}}$ of size 512 and $512 \times 512$, respectively. Each decoder consists of two dense layers of 512 neurons with ReLu activations. The output of decoder 1, 2 and 12 is processed, each, by a fully connected layer without activation to generate $\boldsymbol{\mu}_k^{\mathrm{d}}$ and $\boldsymbol{\Sigma}_k^{\mathrm{d}}$ and $\boldsymbol{\mu}_{12}^{\mathrm{d}}$ and $\boldsymbol{\Sigma}_{12}^{\mathrm{d}}$, of size 2 and $2 \times 2$.

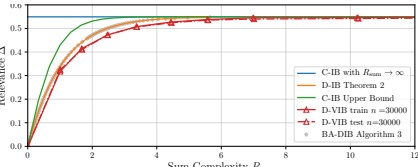 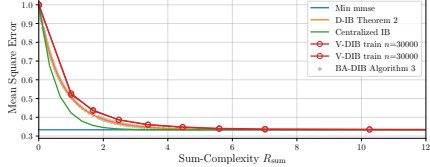

**Figure 5:** Accuracy vs. sum-complexity tradeoffs for the Gaussian data model. Parameters: $K = 2$ encoders, $n_y = 1$, $n_1 = n_2 = 3$ and $n = 30.000$.

**Figure 6:** Mean square error vs. sum-complexity tradeoffs for the Gaussian data model. Parameters: $K = 2$ encoders, $n_y = 1$, $n_1 = n_2 = 3$ and $n = 30.000$.

Figure 5 shows the optimal relevance-complexity region of tuples $(\Delta, R_{\mathrm{sum}})$ obtained from Theorem 2 for a vector Gaussian model with $K = 2$ encoders, target variable dimension $n_y = 1$, and observations dimension $n_1 = n_2 = 3$. A set of 40.000 samples split among training (30.000 samples) and test (10.000 samples). The figure depicts all accuracy-complexity pairs obtained by application of our algorithm D-VIB to this setting. The results are compared to the case of inference with known joint distribution (referred to as D-IB, see next section) as well as the case of centralized inference (C-IB). For the D-VIB algorithm, the the DNN architecture for the coders is shown in Table 6.2. Figure 6 shows the evolution of the associated mean squared error (MSE) in the estimation of the label $Y$ using our D-VIB algorithm. As it can bee seen from both figures the performance of our D-VIB algorithm (which does not require knowledge of the joint label-feature distribution) is very close to that predicted by the theory, i.e., our Theorem 2.

Figure 7 shows similar curves for $n_y = 2$, $n_1 = n_2 = 3$ dimensions, for various sizes of the training datset. As expected large training sets allow a more accurate prediction. Noteworthy, that the performance during the training phase might be better than that of the centralized learning scenario is an indicator can be caused by overfitting. Related to this aspect, recall that although the D-VIB algorithm does not estimate the underlying distribution explicitly, intuitively it does for the computation of the cost function. This is related to that universal compressors also learn the actual distribution of the data that is being compressed. Recall that since the plug-in estimator of entropy is biased downward, estimations of the mutual information terms that are involved in the cost function are then biased upward, which is an alternate explanation to the observed overfitting during the training phase.

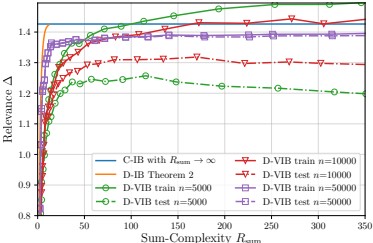

**Figure 7:** Effect of varying training set size $n = \{5.000, 10.000, 50.000\}$ on Gaussian inference for D-VIB.

|  | DNN Layers |
| --- | --- |
| Encoder $k$ | dense [512]-ReLu |
|  | dense [512]-ReLu |
|  | dense [512]-ReLu |
| Lat. space $k$ | dense [256]-ReLu |
| Decoder 12 | dense [256]-ReLu |
| Decoder $k$ | dense [256]-ReLu |

**Table 3:** Used DNN architecture.

## 7 DISTRIBUTED BLAHUT-ARIMOTO TYPE ALGORITHMS

### 7.1 DISCRETE-ALPHABET SETTING

In this section, we derive an iterative method to optimize the variational DIB cost function in equation 23 when the data model is discrete and the joint distribution $P_{X_{\mathcal{K}},Y}$ is either known, or a good estimation of it can be obtained from the training samples. In these cases, the maximizing distributions $\mathbf{P}, \mathbf{Q}$ of the variational DIB cost in equation 23 can be efficiently found by an alternating optimization procedure over $\mathbf{P}$ and $\mathbf{Q}$ similar to the expectation-maximization (EM) algorithm Dempster et al. (1977) and the standard Blahut-Arimoto (BA) methodBlahut (1972). An extension to the vector Gaussian data model, which involves random variable with continuous alphabets, is also provided. The main idea of the algorithm is that at iteration $t$, the optimal distributions $\mathbf{P}^{(t)}$ that maximize the variational D-IB bound $\mathcal{L}_s^{\mathrm{VB}}(\mathbf{P}, \mathbf{Q}^{(t)})$ for fixed $\mathbf{Q}^{(t)}$ can be optimized in closed form and, next, the maximizing pmfs $\mathbf{Q}^{(t)}$ for given $\mathbf{P}^{(t)}$ can be also found analytically. So, starting from an initialization $\mathbf{P}^{(0)}$ and $\mathbf{Q}^{(0)}$ the algorithms performs the following computations successively and in this order, until convergence,

$$\mathbf{P}^{(0)} \to \mathbf{Q}^{(0)} \to \mathbf{P}^{(1)} \to \ldots \to \mathbf{P}^{(t)} \to \mathbf{Q}^{(t)} \to \ldots \tag{81}$$

We refer to such algorithm as "Blahut-Arimoto Distributed Information Bottleneck Algorithm (BA-DIB)". Algorithm 1 describes the steps taken by BA-DIB to successively maximize $\mathcal{L}_s^{\mathrm{VB}}(\mathbf{P}, \mathbf{Q})$ by solving a concave optimization problem over $\mathbf{P}$ and over $\mathbf{Q}$ at each iteration. We have the following lemma whose proof follows essentially by using the log-sum inequality Cover and Thomas (1991) and the convexity of the mapping $x \mapsto x \log x$.

**Lemma 4** *The function $\mathcal{L}_s^{\mathrm{VB}}(\mathbf{P}, \mathbf{Q})$ is concave in $\mathbf{P}$ and in $\mathbf{Q}$.*

For fixed $\mathbf{P}^{(t)}$, the optimal $\mathbf{Q}^{(t)}$ maximizing the variational D-IB bound in equation 19 follows from Lemma 1 as given by equation 21-equation 22. For fixed $\mathbf{Q}^{(t)}$, the optimal $\mathbf{P}^{(t)}$ can be found using the following lemma.

**Lemma 5** *For fixed $\mathbf{Q}$, there exists a $\mathbf{P}$ that achieves the maximum $\max_{\mathbf{P}} \mathcal{L}_s^{\mathrm{VB}}(\mathbf{P}, \mathbf{Q})$, where $P_{U_k|X_k}$ is given by*

$$p^*(u_k|x_k) = q(u_k) \frac{\exp\left(-\psi_s(u_k, x_k)\right)}{\sum_{u_k \in \mathcal{U}_k} q(u_k) \exp(-\psi_s(u_k, x_k))}, \tag{82}$$

*for $u_k \in \mathcal{U}_k$ and $x_k \in \mathcal{X}_k$, $k \in \mathcal{K}$, and where we define*

$$\psi_s(u_k, x_k) := D_{\mathrm{KL}}(P_{Y|x_k}||Q_{Y|u_k}) + \frac{1}{s}\mathbb{E}_{U_{\mathcal{K}\setminus k}|x_k}[D_{\mathrm{KL}}(P_{Y|U_{\mathcal{K}\setminus k},x_k}||Q_{Y|U_{\mathcal{K}\setminus k},u_k})]. \tag{83}$$

**Proof:** Due to its concavity, to maximize $\mathcal{L}_s^{\mathrm{VB}}(\mathbf{P}, \mathbf{Q})$ with respect to $\mathbf{P}$ for given $\mathbf{Q}$, we add the Lagrange multipliers $\lambda_{x_k} \geq 0$ for each constraint $\sum_{u_k \in \mathcal{U}_k} p(u_k|x_k) = 1$ with $x_k \in \mathcal{X}_k$. For each $s$, $\lambda_{x_k} \geq 0$ and $p(u_k|x_k)$ can be explicitly found by solving the KKT conditions, e.g.,

$$\frac{\partial}{\partial p(u_k|x_k)}\left[\mathcal{L}_s^{\mathrm{VB}}(\mathbf{P}, \mathbf{Q}) + \sum_{x_k \in \mathcal{X}_k} \lambda_{x_k}\left(\sum_{u_k \in \mathcal{U}_k} p(u_k|x_k) - 1\right)\right] = 0.$$

This completes the proof.

---

**Algorithm 1** BA-DIB training algorithm for discrete data

---

1: **inputs:**
  discrete pmf $P_{X_1,\ldots,X_k,Y}$, parameter $s \geq 0$.
2: **output:** optimal $P^*_{U_k|X_k}$, pair $(\Delta_s, R_s)$.
3: **initialization**
  Set $t = 0$ and set $\mathbf{P}^{(0)}$ with $p(u_k|x_k) = \frac{1}{|\mathcal{U}_k|}$ for $u_k \in \mathcal{U}_k$, $x_k \in \mathcal{X}_k$, $k = 1,\ldots,K$.
4: **repeat**
5: Compute $\mathbf{Q}^{(t+1)}$ using equation 21 and equation 22.
6: Compute $\mathbf{P}^{(t+1)}$ using equation 82.
7: $t \leftarrow t + 1$
8: **until** convergence.

---

### 7.1.1 CONVERGENCE

Algorithm 1 essentially falls into the class of the Successive Upper-Bound Minimization (SUM) algorithms Razaviyayn et al. (2013) in which $\mathcal{L}_s^{\mathrm{VB}}(\mathbf{P}, \mathbf{Q})$ acts as a globally tight lower bound on $\mathcal{L}_s(\mathbf{P})$. Algorithm 1 provides a sequence $\mathbf{P}^{(t)}$ for each iteration $t$, which converges to a stationary point of the optimization problem equation 23.

**Proposition 4** *Every limit point of the sequence $\mathbf{P}^{(t)}$ generated by Algorithm 1 converges to a stationary point of equation 23.*

**Proof:** Let $\mathbf{Q}^*(\mathbf{P}) = \arg\max_{\mathbf{Q}} \mathcal{L}_s^{\mathrm{VB}}(\mathbf{P}, \mathbf{Q})$. Using Lemma 1, for every $\mathbf{P}' \neq \mathbf{P}$, it holds that

$$\mathcal{L}_s^{\mathrm{VB}}(\mathbf{P}, \mathbf{Q}^*(\mathbf{P}')) \leq \mathcal{L}_s^{\mathrm{VB}}(\mathbf{P}, \mathbf{Q}^*(\mathbf{P}))$$
$$= \mathcal{L}_s(\mathbf{P}). \tag{84}$$

Since $\mathcal{L}_s(\mathbf{P})$ and $\mathcal{L}_s^{\mathrm{VB}}(\mathbf{P}, \mathbf{Q}^*(\mathbf{P}'))$ satisfy the assumptions of (Razaviyayn et al., 2013, Proposition 1), then $\mathcal{L}_s^{\mathrm{VB}}(\mathbf{P}, \mathbf{Q}^*(\mathbf{P}'))$ satisfies A1-A4 in Razaviyayn et al. (2013). Convergence to a stationary point of equation 23 follows from (Razaviyayn et al., 2013, Theorem 1).

The self consistent equations equation 21, equation 22 and equation 83 satisfied by any stationary point of the D-IB problem extend those of the standard point-to-point IB problem Globerson and Tishby (2004) to the distributed IB problem with $K \geq 2$ encoders. In particular, note the additional divergence term in equation 83.

### 7.2 GAUSSIAN SETTING

Recall Algorithm 1. For finite alphabet sources the updating rules of $\mathbf{Q}^{(t+1)}$ and $\mathbf{P}^{(t+1)}$ in Algorithm 1 are relatively easy, but they become unfeasible for continuous alphabet sources. We leverage on the optimality of Gaussian test channels, shown in Theorem 2, to restrict the optimization of $\mathbf{P}$ to Gaussian distributions, which are easily represented by a finite set of parameters, namely mean and covariance. We show that if $\mathbf{P}^{(t)}$ are Gaussian distributions, then $\mathbf{P}^{(t+1)}$ are also Gaussian distributions, which can be computed with an efficient update algorithm of its representing parameters. In particular, if at time $t$ the $k$-th distributions $P_{\mathbf{U}_k|\mathbf{X}_k}^{(t)}$ is given by

$$\mathbf{U}_k^t = \mathbf{A}_k^t \mathbf{X}_k + \mathbf{Z}_k^t, \tag{85}$$

where $\mathbf{Z}_k^t \sim \mathcal{CN}(\mathbf{0}, \boldsymbol{\Sigma}_{\mathbf{z}_k^t})$, we show that at $t+1$, for $\mathbf{P}^{(t+1)}$ updated as in equation 82, the encoder $P_{\mathbf{U}_k|\mathbf{X}_k}^{(t+1)}$ corresponds to $\mathbf{U}_k^{t+1} = \mathbf{A}_k^{t+1} \mathbf{X}_k + \mathbf{Z}_k^{t+1}$, where $\mathbf{Z}_k^{t+1} \sim \mathcal{CN}(\mathbf{0}, \boldsymbol{\Sigma}_{\mathbf{z}_k^{t+1}})$ and $\boldsymbol{\Sigma}_{\mathbf{z}_k^{t+1}}, \mathbf{A}_k^{t+1}$ are updated as

$$\boldsymbol{\Sigma}_{\mathbf{z}_k^{t+1}} = \left( \left(1 + \frac{1}{s}\right) \boldsymbol{\Sigma}_{\mathbf{u}_k^t|\mathbf{y}}^{-1} - \frac{1}{s} \boldsymbol{\Sigma}_{\mathbf{u}_k^t|\mathbf{u}_{\mathcal{K}\backslash k}^t}^{-1} \right)^{-1}, \tag{86}$$

$$\mathbf{A}_k^{t+1} = \boldsymbol{\Sigma}_{\mathbf{z}_k^{t+1}} \left( \left(1 + \frac{1}{s}\right) \boldsymbol{\Sigma}_{\mathbf{u}_k^t|\mathbf{y}}^{-1} \mathbf{A}_k^t (\mathbf{I} - \boldsymbol{\Sigma}_{\mathbf{x}_k|\mathbf{y}} \boldsymbol{\Sigma}_{\mathbf{x}_k}^{-1}) - \frac{1}{s} \boldsymbol{\Sigma}_{\mathbf{u}_k^t|\mathbf{u}_{\mathcal{K}\backslash k}^t}^{-1} \mathbf{A}_k^t (\mathbf{I} - \boldsymbol{\Sigma}_{\mathbf{x}_k|\mathbf{u}_{\mathcal{K}\backslash k}^t} \boldsymbol{\Sigma}_{\mathbf{x}_k}^{-1}) \right). \tag{87}$$

The detailed update procedure is given in Algorithm 2 (see the following section for the details of the derivations).

---

**Algorithm 2** BA-DIB algorithm for the Gaussin Vector D-IB

---

1: **inputs:**
    covariance $\boldsymbol{\Sigma}_{\mathbf{y},\mathbf{x}_1,\ldots,\mathbf{x}_k}$, parameter $s \geq 0$.
2: **output:** optimal pairs $(\mathbf{A}_k^*, \boldsymbol{\Sigma}_{\mathbf{z}_k^*})$, $k = 1, \ldots, K$.
3: **initialization**
    Set randomly $\mathbf{A}_k^0$ and $\boldsymbol{\Sigma}_{\mathbf{z}_k^0} \succeq 0$, $k \in \mathcal{K}$.
4: **repeat**
5: Compute $\boldsymbol{\Sigma}_{\mathbf{x}_k | \mathbf{u}_{\mathcal{K}\setminus k}^t}$ and update for $k \in \mathcal{K}$

$$\boldsymbol{\Sigma}_{\mathbf{u}_k^t | \mathbf{y}} = \mathbf{A}_k^t \boldsymbol{\Sigma}_{\mathbf{x}_k | \mathbf{y}} \mathbf{A}_k^{t,\dagger} + \boldsymbol{\Sigma}_{\mathbf{z}_k^t} \tag{88}$$

$$\boldsymbol{\Sigma}_{\mathbf{u}_k^t | \mathbf{u}_{\mathcal{K}\setminus k}^t} = \mathbf{A}_k^t \boldsymbol{\Sigma}_{\mathbf{x}_k | \mathbf{u}_{\mathcal{K}\setminus k}^t} \mathbf{A}_k^{t,\dagger} + \boldsymbol{\Sigma}_{\mathbf{z}_k^t}, \tag{89}$$

6: Compute $\boldsymbol{\Sigma}_{\mathbf{z}_k^{t+1}}$ as in equation 86 for $k \in \mathcal{K}$.
7: Compute $\mathbf{A}_k^{t+1}$ as equation 87, $k \in \mathcal{K}$.
8: $t \leftarrow t + 1$.
9: **until** convergence.

---

### 7.2.1 DERIVATION OF ALGORITHM 2

We derive the update rules of Algorithm 2 and show that the Gaussian distribution is invariant to the update rules in Algorithm 1, in line with Theorem 2. First, we recall that if $(\mathbf{X}_1, \mathbf{X}_2)$ are jointly Gaussian, then

$$P_{\mathbf{X}_2|\mathbf{X}_1=\mathbf{x}_1} = \mathcal{CN}(\boldsymbol{\mu}_{\mathbf{x}_2|\mathbf{x}_1}, \boldsymbol{\Sigma}_{\mathbf{x}_2|\mathbf{x}_1}), \tag{90}$$

where $\boldsymbol{\mu}_{\mathbf{x}_2|\mathbf{x}_1} := \mathbf{K}_{\mathbf{x}_2|\mathbf{x}_1}\mathbf{x}_1$, with $\mathbf{K}_{\mathbf{x}_2|\mathbf{x}_1} := \boldsymbol{\Sigma}_{\mathbf{x}_2,\mathbf{x}_1}\boldsymbol{\Sigma}_{\mathbf{x}_1}^{-1}$.

Then, for $\mathbf{Q}^{(t+1)}$ computed as in equation 21 and equation 22 from $\mathbf{P}^{(t)}$, which is a set of Gaussian distributions, we have

$$Q_{\mathbf{Y}|\mathbf{u}_k}^{(t+1)} = \mathcal{CN}(\boldsymbol{\mu}_{\mathbf{y}|\mathbf{u}_k^t}, \boldsymbol{\Sigma}_{\mathbf{y}|\mathbf{u}_k^t}),$$

$$Q_{\mathbf{Y}|\mathbf{u}_{\mathcal{K}}}^{(t+1)} = \mathcal{CN}(\boldsymbol{\mu}_{\mathbf{y}|\mathbf{u}_{\mathcal{K}}^t}, \boldsymbol{\Sigma}_{\mathbf{y}|\mathbf{u}_{\mathcal{K}}^t}).$$

Next, we look at the update $\mathbf{P}^{(t+1)}$ as in equation 82 from given $\mathbf{Q}^{(t+1)}$. First, we have that $p(\mathbf{u}_k^t)$ is the marginal of $\mathbf{U}_k^t$, given by $\mathbf{U}_k^t \sim \mathcal{CN}(\mathbf{0}, \boldsymbol{\Sigma}_{\mathbf{u}_k^t})$ where $\boldsymbol{\Sigma}_{\mathbf{u}_k^t} = \mathbf{A}_k^t \boldsymbol{\Sigma}_{\mathbf{x}_k} \mathbf{A}_k^{t,H} + \boldsymbol{\Sigma}_{\mathbf{z}_k^t}$.

Then, to compute $\psi_s(\mathbf{u}_k^t, \mathbf{x}_k)$, first, we note that

$$E_{U_{\mathcal{K}\setminus k}|x_k}[D_{\mathrm{KL}}(P_{Y|U_{\mathcal{K}\setminus k},x_k}||Q_{Y|U_{\mathcal{K}\setminus k},u_k})] = D_{\mathrm{KL}}(P_{Y,U_{\mathcal{K}\setminus k}|x_k}||Q_{Y,U_{\mathcal{K}\setminus k}|u_k}) - D_{\mathrm{KL}}(P_{U_{\mathcal{K}\setminus k}|x_k}||Q_{U_{\mathcal{K}\setminus k}|u_k}) \tag{91}$$

and that for two generic multivariate Gaussian distributions $P_1 \sim \mathcal{CN}(\boldsymbol{\mu}_1, \boldsymbol{\Sigma}_1)$ and $P_2 \sim \mathcal{CN}(\boldsymbol{\mu}_2, \boldsymbol{\Sigma}_2)$ in $\mathbb{C}^N$, the KL divergence is computed as in equation 80 below.

Applying equation 91 and equation 80 in equation 83 and noting that all involved distributions are Gaussian, it follows that $\psi_s(\mathbf{u}_k^t, \mathbf{x}_k)$ is a quadratic form. Then, since $p(\mathbf{u}_k^t)$ is Gaussian, the product $\log(p(\mathbf{u}_k^t)\exp(-\psi_s(\mathbf{u}_k^t, \mathbf{x}_k)))$ is also a quadratic form, and identifying constant, first and second order terms, we can write

$$\log p^{(t+1)}(\mathbf{u}_k|\mathbf{x}_k) = Z(\mathbf{x}_k) + (\mathbf{u}_k - \boldsymbol{\mu}_{\mathbf{u}_k^{t+1}|\mathbf{x}_k})^H \boldsymbol{\Sigma}_{\mathbf{z}_k^{t+1}}^{-1}(\mathbf{u}_k - \boldsymbol{\mu}_{\mathbf{u}_k^{t+1}|\mathbf{x}_k}), \tag{92}$$

where $Z(\mathbf{x}_k)$ is a normalization term independent of $\mathbf{u}_k$,

$$\boldsymbol{\Sigma}_{\mathbf{z}_k^{t+1}}^{-1} = \boldsymbol{\Sigma}_{\mathbf{u}_k^t}^{-1} + \mathbf{K}_{\mathbf{y}|\mathbf{u}_k^t}^H \boldsymbol{\Sigma}_{\mathbf{y}|\mathbf{u}_k^t}^{-1} \mathbf{K}_{\mathbf{y}|\mathbf{u}_k^t}$$
$$+ \frac{1}{s}\mathbf{K}_{\mathbf{yu}_{\mathcal{K}\setminus k}^t|\mathbf{u}_k^t}^H \boldsymbol{\Sigma}_{\mathbf{yu}_{\mathcal{K}\setminus k}^t|\mathbf{u}_k^t}^{-1} \mathbf{K}_{\mathbf{yu}_{\mathcal{K}\setminus k}^t|\mathbf{u}_k^t} - \frac{1}{s}\mathbf{K}_{\mathbf{u}_{\mathcal{K}\setminus k}^t|\mathbf{u}_k^t}^H \boldsymbol{\Sigma}_{\mathbf{u}_{\mathcal{K}\setminus k}^t|\mathbf{u}_k^t}^{-1} \mathbf{K}_{\mathbf{u}_{\mathcal{K}\setminus k}^t|\mathbf{u}_k^t}, \tag{93}$$

and

$$\boldsymbol{\mu}_{\mathbf{u}_k^{t+1}|\mathbf{x}_k} = \boldsymbol{\Sigma}_{\mathbf{z}_k^{t+1}}\left(\mathbf{K}_{\mathbf{y}|\mathbf{u}_k^t}^H \boldsymbol{\Sigma}_{\mathbf{y}|\mathbf{u}_k^t}^{-1} \boldsymbol{\mu}_{\mathbf{y}|\mathbf{x}_k}\right.$$

$$+ \frac{1}{s} \mathbf{K}_{\mathbf{y}, \mathbf{u}^t_{\mathcal{K} \backslash k} | \mathbf{u}^t_k} \mathbf{\Sigma}^{-1}_{\mathbf{y}, \mathbf{u}^t_{\mathcal{K} \backslash k} | \mathbf{u}^t_k} \boldsymbol{\mu}_{\mathbf{y}, \mathbf{u}^t_{\mathcal{K} \backslash k} | \mathbf{x}_k} - \frac{1}{s} \mathbf{K}_{\mathbf{u}^t_{\mathcal{K} \backslash k} | \mathbf{u}^t_k} \mathbf{\Sigma}^{-1}_{\mathbf{u}^t_{\mathcal{K} \backslash k} | \mathbf{u}^t_k} \boldsymbol{\mu}_{\mathbf{u}^t_{\mathcal{K} \backslash k} | \mathbf{x}_k} \Bigg) . \tag{94}$$

This shows that $p^{(t+1)}(\mathbf{u}_k | \mathbf{x}_k)$ is a multivariate Gaussian distribution and that $\mathbf{U}^{t+1}_k | \{ \mathbf{X}_k = \mathbf{x}_k \}$ is also a multivariate Gaussian distributed as $\mathcal{CN}(\boldsymbol{\mu}_{\mathbf{u}^{t+1}_k | \mathbf{x}_k}, \mathbf{\Sigma}_{\mathbf{z}^{t+1}_k})$.

Next, we simplify equation 93 and equation 94 to obtain the update rules equation 86 and equation 87. From the matrix inversion lemma, similarly to Chechik et al. (Feb. 2005), for $(\mathbf{X}_1, \mathbf{X}_2)$ jointly Gaussian we have

$$\mathbf{\Sigma}^{-1}_{\mathbf{x_2} | \mathbf{x_1}} = \mathbf{\Sigma}^{-1}_{\mathbf{x_2}} + \mathbf{K}^H_{\mathbf{x_1} | \mathbf{x_2}} \mathbf{\Sigma}^{-1}_{\mathbf{x_1} | \mathbf{x_2}} \mathbf{K}_{\mathbf{x_1} | \mathbf{x_2}}. \tag{95}$$

Applying equation 95, in equation 93 we have

$$\mathbf{\Sigma}^{-1}_{\mathbf{z}^{t+1}_k} = \mathbf{\Sigma}^{-1}_{\mathbf{u}^t_k | \mathbf{y}} + \frac{1}{s} \mathbf{\Sigma}^{-1}_{\mathbf{u}^t_k | \mathbf{y} \mathbf{u}^t_{\mathcal{K} \backslash k}} - \frac{1}{s} \mathbf{\Sigma}^{-1}_{\mathbf{u}^t_k | \mathbf{u}^t_{\mathcal{K} \backslash k}}, \tag{96}$$

$$= \left( 1 + \frac{1}{s} \right) \mathbf{\Sigma}^{-1}_{\mathbf{u}^t_k | \mathbf{y}} - \frac{1}{s} \mathbf{\Sigma}^{-1}_{\mathbf{u}^t_k | \mathbf{u}^t_{\mathcal{K} \backslash k}}, \tag{97}$$

where equation 97 is due to the Markov chain $\mathbf{U}_k \; \multimap \; \mathbf{Y} \; \multimap \; \mathbf{U}_{\mathcal{K} \backslash k}$.

Then, also from the matrix inversion lemma, we have for jointly Gaussian $(\mathbf{X}_1, \mathbf{X}_2)$,

$$\mathbf{\Sigma}^{-1}_{\mathbf{x_2} | \mathbf{x_1}} \mathbf{\Sigma}_{\mathbf{x_2}, \mathbf{x_1}} \mathbf{\Sigma}^{-1}_{\mathbf{x_1}} = \mathbf{\Sigma}^{-1}_{\mathbf{x_2}} \mathbf{\Sigma}_{\mathbf{x_2}, \mathbf{x_1}} \mathbf{\Sigma}^{-1}_{\mathbf{x_1} | \mathbf{x_2}}. \tag{98}$$

Applying equation 98 to equation 94, for the first term in equation 94, we have

$$\mathbf{K}^H_{\mathbf{y} | \mathbf{u}^t_k} \mathbf{\Sigma}^{-1}_{\mathbf{y} | \mathbf{u}^t_k} \boldsymbol{\mu}_{\mathbf{y} | \mathbf{x}_k} = \mathbf{\Sigma}^{-1}_{\mathbf{u}^t_k | \mathbf{y}} \mathbf{\Sigma}_{\mathbf{y}, \mathbf{u}^t_k} \mathbf{\Sigma}^{-1}_{\mathbf{y}} \boldsymbol{\mu}_{\mathbf{y} | \mathbf{x}_k} \tag{99}$$

$$= \mathbf{\Sigma}^{-1}_{\mathbf{u}^t_k | \mathbf{y}} \mathbf{A}^t_k \mathbf{\Sigma}_{\mathbf{x}_k, \mathbf{y}} \mathbf{\Sigma}^{-1}_{\mathbf{y}} \mathbf{\Sigma}_{\mathbf{y}, \mathbf{x}_k} \mathbf{\Sigma}^{-1}_{\mathbf{x}_k} \mathbf{x}_k$$

$$= \mathbf{\Sigma}^{-1}_{\mathbf{u}^t_k | \mathbf{y}} \mathbf{A}^t_k (\mathbf{I} - \mathbf{\Sigma}_{\mathbf{x}_k | \mathbf{y}} \mathbf{\Sigma}^{-1}_{\mathbf{x}_k}) \mathbf{x}_k, \tag{100}$$

where $\mathbf{\Sigma}_{\mathbf{y}, \mathbf{u}^t_k} = \mathbf{A}^t_k \mathbf{\Sigma}_{\mathbf{x}_k, \mathbf{y}}$; and equation 100 is due to the definition of $\mathbf{\Sigma}_{\mathbf{x}_k | \mathbf{y}}$.

Similarly, for the second term in equation 94, we have

$$\mathbf{K}_{\mathbf{y} \mathbf{u}^t_{\mathcal{K} \backslash k} | \mathbf{u}^t_k} \mathbf{\Sigma}^{-1}_{\mathbf{y} \mathbf{u}^t_{\mathcal{K} \backslash k} | \mathbf{u}^t_k} \boldsymbol{\mu}_{\mathbf{y}, \mathbf{u}^t_{\mathcal{K} \backslash k} | \mathbf{x}_k} = \mathbf{\Sigma}^{-1}_{\mathbf{u}^t_k | \mathbf{y} \mathbf{u}^t_{\mathcal{K} \backslash k}} \mathbf{A}^t_k (\mathbf{I} - \mathbf{\Sigma}_{\mathbf{x}_k | \mathbf{y} \mathbf{u}^t_{\mathcal{K} \backslash k}} \mathbf{\Sigma}^{-1}_{\mathbf{x}_k}) \mathbf{x}_k, \tag{101}$$

$$= \mathbf{\Sigma}^{-1}_{\mathbf{u}^t_k | \mathbf{y}} \mathbf{A}^t_k (\mathbf{I} - \mathbf{\Sigma}_{\mathbf{x}_k | \mathbf{y}} \mathbf{\Sigma}^{-1}_{\mathbf{x}_k}) \mathbf{x}_k, \tag{102}$$

where we use $\mathbf{\Sigma}_{\mathbf{u}^t_k, \mathbf{y} \mathbf{u}^t_{\mathcal{K} \backslash k}} = \mathbf{A}^t_k \mathbf{\Sigma}_{\mathbf{x}_k, \mathbf{y} \mathbf{u}^t_{\mathcal{K} \backslash k}}$; and equation 102 is due to the Markov chain $\mathbf{U}_k \; \multimap \; \mathbf{Y} \; \multimap \; \mathbf{U}_{\mathcal{K} \backslash k}$.

For the third term in equation 94,

$$\mathbf{K}_{\mathbf{u}^t_{\mathcal{K} \backslash k} | \mathbf{u}^t_k} \mathbf{\Sigma}^{-1}_{\mathbf{u}^t_{\mathcal{K} \backslash k} | \mathbf{u}^t_k} \boldsymbol{\mu}_{\mathbf{u}^t_{\mathcal{K} \backslash k} | \mathbf{x}_k} = \mathbf{\Sigma}^{-1}_{\mathbf{u}^t_k | \mathbf{u}^t_{\mathcal{K} \backslash k}} \mathbf{A}^t_k (\mathbf{I} - \mathbf{\Sigma}_{\mathbf{x}_k | \mathbf{u}^t_{\mathcal{K} \backslash k}} \mathbf{\Sigma}^{-1}_{\mathbf{x}_k}) \mathbf{x}_k. \tag{103}$$

Equation equation 87 follows by noting that $\boldsymbol{\mu}_{\mathbf{u}^{t+1}_k | \mathbf{x}_k} = \mathbf{A}^{t+1}_k \mathbf{x}_k$, and that from equation 94 $\mathbf{A}^{t+1}_k$ can be identified as in equation 87.

Finally, we note that due to equation 85, $\mathbf{\Sigma}_{\mathbf{u}^t_k | \mathbf{y}}$ and $\mathbf{\Sigma}_{\mathbf{u}^t_k | \mathbf{u}^t_{\mathcal{K} \backslash k}}$ are given as in equation 88 and equation 89, where $\mathbf{\Sigma}_{\mathbf{x}_k | \mathbf{y}} = \mathbf{\Sigma}_k$ and $\mathbf{\Sigma}_{\mathbf{x}_k | \mathbf{u}^t_{\mathcal{K} \backslash k}}$ can be computed from its definition. This completes the proof.

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
