# OpenReview forum: "An Information Theoretic Approach to Distributed Representation Learning"
_ICLR.cc/2020/Conference — Reject_

### Official Review · AnonReviewer3 · 2019-10-23
**Official Blind Review #3**

**Rating:** 3

**Review:**

The paper extended the Gaussian Information Bottleneck method to the case of multi-view learning and provided a variation bound for the accuracy optimization with constrain on the sum of complexity. It also proposed an algorithm to learn the distributed representation without any prior knowledge of the data distribution.

The multi-view learning problem has been quite well studied in the literature. The paper reformulated the multi-view learning problem as a Bayesian inference problem and provided solid analysis for it.

The writing of the paper was pretty hard to follow for me, with a lot of notations that are not defined clearly.
* For example, I can roughly guess that U in theorem 1 represent the learned descriptors, but what’s the variable T in theorem 1?
* What is \Omega in Theorem 2?

The experimental result doesn’t look very comprehensive at all as it was mostly compared with variations of the proposed algorithm and it doesn’t include any other multi-view learning algorithms.

The algorithms in the experimental result are not very clearly defined. I don’t see much explanation of what is exactly D-VIB and C-VIB. There’s some formulation of the algorithm in Section 3.4, but it only gives a loss function briefly. I’m not sure if many practitioners will be able to implement this algorithm from the description here.


**Experience Assessment:**

I do not know much about this area.

**Review Assessment: Checking Correctness Of Derivations And Theory:**

I assessed the sensibility of the derivations and theory.

**Review Assessment: Checking Correctness Of Experiments:**

I assessed the sensibility of the experiments.

**Review Assessment: Thoroughness In Paper Reading:**

I read the paper at least twice and used my best judgement in assessing the paper.

---

### Official Review · AnonReviewer1 · 2019-10-23
**Official Blind Review #1**

**Rating:** 8

**Review:**

In this paper, the authors studied the distributed representation learning problem, where multiple sources of data are processed to provide information about Y. They studied this problem from information-theoretic point of view. Their main contribution can be summarized as follows.
    1. The optimal trade-off between the accuracy and complexity were studied for discrete memoryless data model as well as memoryless vector Gaussian model.
    2. A variational bound were constructed in order to connect the optimal encoder and decoder mappings with the solution of an optimization algorithm.
    3. If only samples from an unknown distribution are available, an algorithm were proposed to find the optimal encode and decoder. Moreover, some experiment were conducted to support the approach.

In general, I think the paper is well-organized. The definition of the problem and the motivation of the approach are clear. The theorems, algorithms and experiments are solid enough to support the whole story of this paper. Generally I wish to see  this paper being accepted.

**Experience Assessment:**

I have read many papers in this area.

**Review Assessment: Checking Correctness Of Derivations And Theory:**

I did not assess the derivations or theory.

**Review Assessment: Checking Correctness Of Experiments:**

I did not assess the experiments.

**Review Assessment: Thoroughness In Paper Reading:**

I read the paper at least twice and used my best judgement in assessing the paper.

---

### Official Review · AnonReviewer2 · 2019-10-28
**Official Blind Review #2**

**Rating:** 6

**Review:**

I am not an expert in this area and the paper involves a lot of derivations and proofs, but I did not check the correctness of those derivations. In summary, this paper proposed a framework for integrating multiple data sources for representing data. In the framework, each data source was mapped to a latent data variable by using a nonlinear function which is called an encoder; then the mapped latent variables were jointly mapped to the target data by using another nonlinear function which is called a decoder. To make this idea to work, the paper used mutual information as the objective function to control the accuracy if the model, and at the same time to avoid overfitting the paper proposed to use MDL as a measure to control the complexity of the model. If I was right, this was the whole picture of the proposed model. My questions are the following:
1) I am not very clear how the model complexity was automatically incorporated with the objective function. It seems to me that the objective function was finally the equation (29) and then the neural networks for encoder and decoder were optimized. If this was the case, how the model complexity was incorporated, that is, how the R_k was used in the model? Was the values R_k constant in the model - I mean they are fixed constant values? How these values,i.e.,R_k, were chosen?
2) I am a mathematician, but to be honest, I feel that the Maths in the paper is huge and heavy and I thought it could not be that complex for the model. The consequence is that it make the paper to be hard to read. This is a personal feeling, you could just ignore this point.
3) Experiments: there are a lot of papers describing to integrate data sources for at least the MNIST example. It would be interesting to compare the proposed method to the literature. The experiment in 4.1 obviously is a toy data problem - I mean although the data is real, but the data generated was using noisy and rotations. It would be more interesting to apply the method to a real-world problem.
4) I think it would be more friendly to explicitly define the concepts of Discrete Memoryless and Memoryless Vector Gaussian Models.
5) The Markov chain represented in equation (3) is not well defined. I do not understand these notations.
6) Before the equation (4), is it equivalent X_k^n and X_{k,n}? I am confused by these notations
7) In equation (6), it is more readable to explicitly define the Shannon Mutual Information.
8) The second paragraph on Page 5: you use Gaussian pmfs here, but pmf denotes discrete variable. But Gaussian I assume is continuous.

**Experience Assessment:**

I do not know much about this area.

**Review Assessment: Checking Correctness Of Derivations And Theory:**

I assessed the sensibility of the derivations and theory.

**Review Assessment: Checking Correctness Of Experiments:**

I assessed the sensibility of the experiments.

**Review Assessment: Thoroughness In Paper Reading:**

I read the paper at least twice and used my best judgement in assessing the paper.

---

### Official Review · AnonReviewer4 · 2019-11-05
**Official Blind Review #4**

**Rating:** 3

**Review:**

This paper studies a distributed representation problem where multiple features X_1,...,X_K are processed (or encoded) separately to estimate (or decode) some quantity of interest Y.
The log loss is considered throughout, which amounts to measuring the mutual information between Y and \hat Y, defined as the "accuracy" of the estimation method. The average rate (measured in number of bits per sample) of the encoded feature is defined as the "complexity" of the representation method.
The author derived the fundamental trade-off between the accuracy and the complexity for any representation-estimation (or encoding-decoding) method.
The author also derived a variational representation of the optimal accuracy-complexity region, which also expresses the optimal encoder and decoder map as the solution of the optimization problem.
Finally, the author considered the case where the joint distribution of P_{X_1,,,,X_K,Y} is unknown, and encoder and decoder are parameterized by neural networks, parameters of which are tuned using data.

I incline to reject the paper, for the following reasons.
1. The accuracy-complexity trade-off studied in the paper is more of a rate-distortion type of information-theoretic problem, where the joint distribution P_{X_1,,,,X_K,Y} is assumed to be known. Its connection to the learning problem, where the joint distribution P_{X_1,,,,X_K,Y} is unknown, is unclear. Even if the precise accuracy-complexity region is obtained, it says little about the sample complexity needed by a learning algorithm to achieve this region.
2. Deriving the optimal encoder-decoder mapping from the variational representation of the accuracy-complexity region also requires the joint distribution, which violates the basic assumption of the learning problem.
3. The author did consider the case where the joint distribution is unknown, and the encoder-decoder pair is learned from data. However, this learning problem is somewhat artificial: each encoder only encodes one of the features, but in order to encode optimally, it has to know the entire joint distribution, hence need to access all the features during training. This discrepancy of seeing different components of the data set during training and inference is not well-motivated.
The author mentioned "multi-view learning" at the beginning of the paper. It would be good if the author can elaborate more on this problem in Sec 4 of Experiment Results, and discuss with more detail on how the proposed method solves this problem and how it is different from the existing results, both in terms of the algorithm and the performance.



=================================================
Feedback to authors' reply

I got a better understanding on how the proposed learning algorithms works after reading the authors' reply.
I guess the idea for the case where the joint distribution is unknown is that, for encoding, different nodes uses its own training data (without accessing other nodes' data) to optimize the encoder separately; while for decoding, the master node trains the decoder uses data available to all nodes to estimate the joint distribution.
In this way, the encoders and the decoder jointly optimizes a variational lower bound of the optimal rate region.

If this is the case, I think the proposed method may have some value in practice.
But now the question is how good the variational lower bound is compared to the optimal region, and how well can this variational lower bound be approximated by neural networks and how efficient can the training be done. Without theoretical analysis on these questions, one may only use experiments to assess the performance. From Table 2, it looks like the improvement of the proposed method on the existing method is quite marginal.

In summary, I would like to thank the authors' valuable reply. I encourage the authors to study the gap between the variational lower bound and the optimal region, and maybe do more experiments to find a good use case of the proposed method.

**Experience Assessment:**

I have published one or two papers in this area.

**Review Assessment: Checking Correctness Of Derivations And Theory:**

I did not assess the derivations or theory.

**Review Assessment: Checking Correctness Of Experiments:**

I assessed the sensibility of the experiments.

**Review Assessment: Thoroughness In Paper Reading:**

I read the paper at least twice and used my best judgement in assessing the paper.

---

### Decision · Program_Chairs · 2019-12-19

**Decision:**

Reject

**Comment:**

The authors study generalization in distributed representation learning by describing limits in accuracy and complexity which stem from information theory.

The paper has been controversial, but ultimately the reviewers who provided higher scores presented weaker and fewer arguments. By recruiting an additional reviewer it became clearer that, overall the paper needs a little more work to reach ICLR standards. The main suggestions for improvements have to do with improving clarity in a way that makes the motivation convincing and the practicality more obvious. Boosting the experimental results is a complemental way of increasing convincingness, as argued by reviewers.